# PALMBENCH: A COMPREHENSIVE BENCHMARK OF COMPRESSED LARGE LANGUAGE MODELS ON MOBILE PLATFORMS

Yilong Li[1], Jingyu Liu[1], Hao Zhang[1], M Badri Narayanan[1], Utkarsh Sharma[1], Shuai Zhang[2],

Yijing Zeng[1], Jayaram Raghuram[1], Suman Banerjee[1]

[1]University of Wisconsin – Madison, [2]Amazon Web Services AI, USA

## ABSTRACT

Deploying large language models (LLMs) locally on mobile devices is advantageous in scenarios where transmitting data to remote cloud servers is either undesirable due to privacy concerns or impractical due to network connection. Recent advancements (MLC, 2023a; Gerganov, 2023) have facilitated the local deployment of LLMs. However, local deployment also presents challenges, particularly in balancing quality (generative performance), latency, and throughput within the hardware constraints of mobile devices. In this paper, we introduce our lightweight, *all-in-one automated benchmarking* framework that allows users to evaluate LLMs on mobile devices. We provide a comprehensive benchmark of various popular LLMs with different quantization configurations (both weights and activations) across multiple mobile platforms with varying hardware capabilities. Unlike traditional benchmarks that assess full-scale models on high-end GPU clusters, we focus on evaluating resource efficiency (memory and power consumption) and harmful output for compressed models on mobile devices. Our key observations include: **i)** differences in energy efficiency and throughput across mobile platforms; **ii)** the impact of quantization on memory usage, GPU execution time, and power consumption; and **iii)** accuracy and performance degradation of quantized models compared to their non-quantized counterparts; and **iv)** the frequency of hallucinations and toxic content generated by compressed LLMs on mobile devices.[1]

## 1 INTRODUCTION

Large Language Models (LLMs) such as ChatGPT (OpenAI, 2023), Claude (Anthropic, 2023), and Llama (Touvron et al., 2023a;b;c; Llama, 2024) are powerful generative models that are revolutionizing interactive communication and various natural language processing tasks, including question-answering, document summarization, abstract reasoning, and code auto-completion (e.g., Github Copilot (Github)). LLMs require significant computational and memory resources due to their huge number of parameters (e.g., MT-NLG 530B (Smith et al., 2022)), making them more suitable for running on cloud infrastructures with high-end powerful GPU clusters. While significant attention has been dedicated to cloud-based LLMs, there is a growing need to run LLMs on resource-constrained mobile devices to obtain some key benefits. (1) *Privacy and Security*: Processing user data locally on mobile devices helps protect user privacy and enhances data security. There is also less risk of data breaches or unauthorized access to sensitive information. (2) *No Cloud Reliance*: By running LLMs locally, mobile applications can reduce their dependence on cloud services for language processing tasks. This can lead to cost savings and increased reliability, as the application's functionality is not reliant on the availability and performance of remote servers (Hu et al., 2024). (3) *Offline Access*: By running LLMs on mobile devices, users can access powerful language processing capabilities even when they are not connected (or have unreliable connection) to the Internet.

The rapidly flourishing LLM ecosystem, including various large models, architectures and frameworks, presents both opportunities and challenges for developers and researchers interested in deploying pre-trained LLMs on mobile devices. Existing efforts in on-device LLM inference have primarily focused on model compression and efficient inference techniques, with a strong emphasis

---

[1]Code is available at: `https://github.com/JimmyLi-Network/PalmBench.github.io/`

on deploying models on edge SoC (system-on-a-chip) with GPU and Linux systems (Lin et al., 2024; Gerganov, 2023; MLC, 2023a; Lu et al., 2024; Xu et al., 2024). These approaches aim to achieve a desired quality, latency, and throughput while operating within the constraints of the target platform, *e.g.*, available memory and power limitations. However, the challenges and opportunities of efficiently deploying these large models on mobile platforms, such as smartphones or nano computers (*e.g.*, NVIDIA Jetson Orin Nano), remain largely unexplored.

Current LLM benchmarks primarily target the accuracy of large models on cloud clusters rather than mobile devices (Zheng et al., 2023). Some papers on mobile devices typically examines a limited range of models and platforms, often overlooking performance degradation and hallucination due to quantization (Çöplü et al., 2023; Laskaridis et al., 2024; Murthy et al., 2024).

To bridge this gap, we propose a comprehensive benchmarking framework to evaluate the overall user experience of LLMs on mobile devices. This framework automates the evaluation of various LLMs with different compiler options (e.g., weight and activation quantization) across mobile platforms with diverse hardware capabilities (see Table 1). We systematically evaluate each LLM using a range of metrics across efficiency, accuracy (generative quality relative to non-quantized models), and harmful output dimensions. Our primary focus is on inference efficiency on mobile platforms, evaluating the computation usage (CPU and GPU), latency, throughput, energy consumption, and memory footprint of different models on different platforms (as shown in Figure 1). Along the accuracy dimension, we evaluate quantized models on mobile devices using mainstream datasets like Natural Questions (Kwiatkowski et al., 2019) and SQuAD (Rajpurkar et al., 2016), employing exact match and perfect match metrics to quantify performance degradation and ensure proper functionality for basic use cases. Also, we evaluate the harmful outputs including hallucination and toxicity of LLMs with existing benchmark datasets Li et al. (2023b); Lin et al. (2022); Luong et al. (2024).

Table 1: Summary of LLMs, mobile platforms, and quantization configurations explored by our benchmark.

| | |
|---|---|
| LLMs | LLaMa-2 (Touvron et al., 2023b), LLaMa-3/3.1 (Dubey et al., 2024), RedPajama (Computer, 2023), LLaMa-3.2 (Llama, 2024), Vicuna (Chiang et al., 2023), TinyLlama (Zhang et al., 2024), Qwen2 (Bai et al., 2023), Mistral-7B (Jiang et al., 2023a), Gemma (Team, 2024), Deepseek-R1-7b-qwen-distill (DeepSeek-AI, 2025), Phi2 (Javaheripi et al., 2023), Phi3 (Abdin et al., 2024) |
| Mobile Platforms | Google Pixel 4 / Pixel 5a / Pixel 7, iPhone 15 Pro, iPhone 12 Pro, S22 Ultra, Orange Pi 5 (Pi), Nvidia Jetson Nano (Nano) |
| Quantization | 2-bit (MLC, llama.cpp), 4-bit (MLC, llama.cpp), 5-bit (llama.cpp), 6-bit (llama.cpp), 8-bit (MLC, llama.cpp) (Frantar et al., 2022; Lin et al., 2024; Li et al., 2020; Dettmers & Zettlemoyer, 2023) |

Unlike existing benchmarking efforts for LLM deployment on mobile devices, our study explores and evaluates feasible pre-trained models with the most popular quantizations. We highlight various combinations of LLM configurations suitable for mobile deployment (Zhang et al., 2024; Computer, 2023; Team, 2024; Llama, 2024; Abdin et al., 2024), along with several available quantization options (Lin et al., 2024; Frantar et al., 2022; MLC, 2023a; Dettmers & Zettlemoyer, 2023).

We found that 4-bit quantization methods, such as group-wise (Yang et al., 2024), GPTQ (Frantar et al., 2022), and AWQ (Lin et al., 2024), can generally preserve the performance of LLMs while reducing their size to one-quarter of the original non-quantized model. This configuration diversity can potentially lead developers (and users) to make sub-optimal choices in terms of performance and efficiency. Our benchmarking analysis, focused on resource usage during inference, provides insights into efficient deployment strategies tailored for mobile platforms. These strategies include joint considerations of 1) model architecture, 2) quantization strategy, and 3) model size. In summary, the major contributions of our paper are:

- To enable a comprehensive evaluation of various LLMs, we first develop a *lightweight, automated benchmarking framework* that collects performance metrics from mobile devices via USB, eliminating the need for additional equipment.
- We evaluate various quantized LLMs on mobile platforms with different hardware capabilities, measuring their resource utilization, power consumption, throughput, and inference latency.

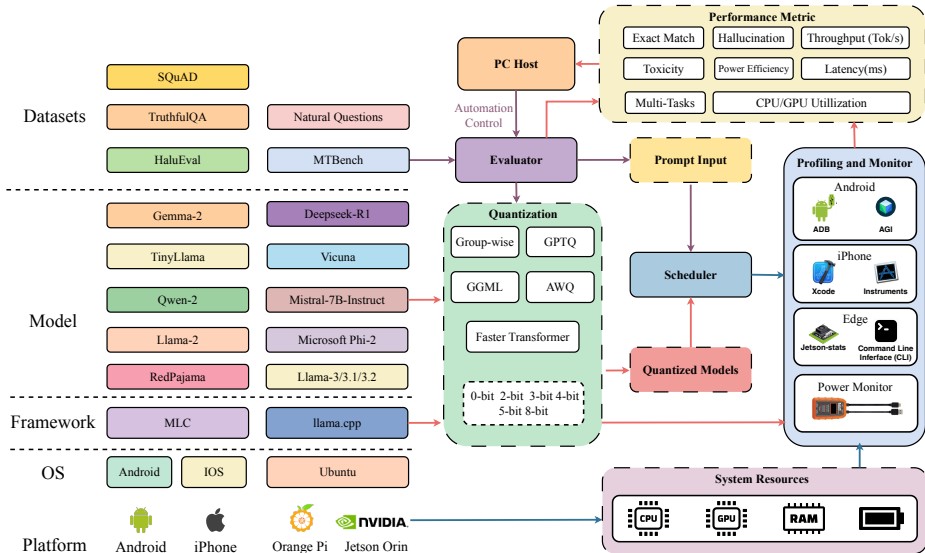

Figure 1: Overview and workflow of `PalmBench` – our evaluation and benchmarking framework for Large Language Models (LLMs) on mobile devices.

- We validate the *knowledge and answering* accuracy of quantized models compared to their non-quantized counterparts, and investigate the potential issues of compressed models such as toxicity, bias, and the generation of erroneous or random outputs (hallucinations).

- Finally, our benchmarking leads to several key observations, highlighting the quantization differences across models, platforms, and frameworks. We also observed that the iOS platform outperforms others in power efficiency, latency, and throughput for LLM inference [2].

## 2 RELATED WORK

Our benchmarking study extensively evaluated prior efforts focused on optimizing LLMs for mobile devices. These efforts include frameworks (MLC, 2023a;b; Gerganov, 2023), the development of smaller models (Computer, 2023; Abdin et al., 2024; Li et al., 2023c), and model quantization techniques (Frantar et al., 2022).

**Large Language Models.** LLMs like ChatGPT (OpenAI, 2023), the Llama series (Touvron et al., 2023a;c; Dubey et al., 2024; Llama, 2024), Mistral (Jiang et al., 2023a), Vicuna (Chiang et al., 2023), Gemma (Team, 2024) *etc.* are gaining substantial influence in generative AI applications. Their resource requirements—both in terms of power consumption and memory usage—scale linearly with model size, introducing significant operational overhead during inference. This challenge has catalyzed research into efficient on-device LLM deployment (Xu et al., 2024; Lin et al., 2024; Zhu et al., 2023; Lu et al., 2024), with particular emphasis on model compression and optimization. Recent advances in compact model architectures have yielded promising alternatives, including Google's Gemma-2-2B (Team, 2024), Llama-3.2-1B/3B (Llama, 2024), RedPajama-INCITE-3B (Computer, 2023), Phi-2/Phi-3 (Abdin et al., 2024), and TinyLlama (Zhang et al., 2024).

**Quantization.** Post-training quantization (PTQ) compresses LLMs after full training, creating smaller models optimized for inference. This method enables more efficient storage and faster computation. Group-wise quantization (Yang et al., 2024) partitions neural network weights into groups, quantizing each independently. This approach better aligns with weight distributions, reducing I/O costs and improving performance on mobile platforms. GPTQ (Frantar et al., 2022) further enhances efficiency by compressing LLM weights to 3 or 4 bits compared to the standard 8-bit quantization. Activation-aware Weight Quantization (AWQ) (Lin et al., 2024) identifies and preserves a subset of model weights with larger activation magnitudes, known as salient weights, to minimize quantization loss.

---

[2]We have released the code of our framework to facilitate reproducibility and extensions of our research.

Prior research (Dettmers & Zettlemoyer, 2023; Huang et al., 2024b; Lu et al., 2024; Xu et al., 2024) has highlighted significant performance differences among quantization algorithms, showing that RTN lags behind GPTQ and AWQ.

**Inference Engine.** While numerous efficient inference frameworks exist, MLC-LLM(MLC, 2023a;b) stands out by enabling users to develop, quantize, and deploy LLMs across diverse platforms, including mobile devices and web browsers. Llama.cpp (Gerganov, 2023), written in C++, offers a lightweight, portable alternative to Python-based frameworks. It supports multiple GPU kernels for high-speed processing and focuses on mixed quantization methods, particularly K-Quants. Both frameworks provide pre-compiled models while allowing users to perform custom quantization as needed.

Table 2: Metrics for evaluating the performance of LLMs on mobile devices. Memory usage includes both the model loaded to the memory and the framework program running on devices.

| Metric | Definition |
|---|---|
| CPU Utilization (%) | Percentage of the total processor cycles consumed by LLM |
| GPU Utilization (%) | Percentage of the total GPU computing resource during LLM inference |
| Memory Footprint (GB) | Measurement of main memory used by the LLM application |
| Memory Utilization (%) | Percentage of main memory used by the LLM application |
| Throughput (Tok / s) | Number of output tokens per second generated by the LLM |
| Output Matching | Accuracy degradation of the compressed model relative to the original model |
| Toxicity | Toxicity score calculated on 25k sentences by Perspective API |
| Hallucination (%) | Percentage of erroneous or random outputs not related to the questions |

**Benchmark.** Most existing benchmark frameworks of LLMs focus on maximizing performance across different model architectures and evaluating a model's general world knowledge, question-answering, and reasoning ability (Zheng et al., 2023; Hendrycks et al., 2021; Rajpurkar et al., 2016; Kwiatkowski et al., 2019). Some existing studies on evaluating LLMs for mobile deployment are limited in scope (Murthy et al., 2024; Çöplü et al., 2023; Laskaridis et al., 2024) or evaluating online models instead of on-device inference (Lee et al., 2024). They fail to comprehensively examine resource efficiency and power consumption, which are crucial for mobile deployment. Recent efforts to benchmark edge LLMs include the Starting-kit competition framework (Huang et al., 2024a), built on MLC (MLC, 2023a;b), which facilitates LLM evaluation on Linux-based edge devices.

MELT (Laskaridis et al., 2024) evaluates five LLMs across devices, measuring throughput, power, and Q&A accuracy, but lacks comprehensive analysis of resource utilization, energy efficiency across quantization methods, and layer-wise GPU profiling. Similarly, MobileAIBench (Murthy et al., 2024) offers comprehensive benchmarks for quantized LLMs and LMMs, with a focus on NLP tasks and resource utilization, but its scope is limited to iOS devices and lacks both automated testing capabilities and cross-platform evaluation support. Furthermore, all these existing benchmark works ignore the toxicity, bias, and generation of erroneous or randomized outputs (which always occur in quantized models)—factors that significantly impact user experience across different frameworks.

## 3 METHODOLOGY

To evaluate the LLMs on mobile devices, we created the `PalmBench` framework, which focuses on the following three aspects:

**1) Benchmark Automation.** We designed an automated framework comprising an **Evaluator**, **Scheduler** and **Profiling and Monitor** modules, designed to handle experimental prompt datasets, evaluate quantized models, and schedule profiling across various metrics. The framework's workflow is illustrated in Fig. 1.

**2) Resource Utilization.** Our primary focus is on the resource demands of different models across various platforms—such as CPU, GPU, and memory that significantly impact user experience. Our study goes beyond resource demands, aiming to quantify the impact of different quantization techniques on both performance and resource efficiency across various state-of-the-art models.

**3) Model Accuracy.** Although a model architecture largely dictates its outputs, we observe device-specific variations under different quantization methods. To validate quantized LLMs and evaluate the

compression-induced accuracy drop, we use *Exact Match* and *F1* scores, comparing their *knowledge and answering* accuracy to the original models (calculation methods are detailed in the Appendix A.3). Additionally, we test these models on standard tasks with open-source datasets such as SQuAD and Natural Questions (Rajpurkar et al., 2016; Kwiatkowski et al., 2019). Moreover, we also investigate the potential issues of toxicity and the generation of erroneous or randomized outputs that always occur in compressed models and have not been thoroughly studied in previous work (Laskaridis et al., 2024).

### 3.1 METRICS AND DATASETS

Table 2 summarizes all the metrics used in our benchmark. To evaluate the accuracy and correctness of quantized LLMs, we use popular *Question-Answering* datasets such as SQuAD (Rajpurkar et al., 2016) and Natural Questions (NQ) (Kwiatkowski et al., 2019), comparing their performance with that of the original, non-quantized models. Additionally, we employ comprehensive benchmarks for different tasks, including MTBench (Zheng et al., 2023) to compare their language understanding and reasoning capabilities across different quantizations. Also, we measured *toxicity* by calculating **toxic score** by using Perspective API (Perspective, 2020) and TET (Luong et al., 2024) and evaluated *hallucination* in each quantized LLM using HaluEval (Li et al., 2023a) and TruthfulQA (Lin et al., 2022) benchmarks. The Appendix A.7 and Appendix A.9 provide some examples of these datasets for benchmarking. These widely-recognized datasets ensure that our experiments and metrics are both convincing and reproducible.

### 3.2 CHOICE OF LLMS

We have identified and converted several popular models for benchmarking on edge and mobile devices using model weights from their official Huggingface or GitHub repositories. These models are converted into experimental formats such as GGUF and K-quant for Llama.cpp (Gerganov, 2023) or compiled using TVM (Chen et al., 2018) for MLC frameworks (MLC, 2023a;b). Given that mobile devices typically do not exceed 8GB of memory, it is impractical to test models that are too large, as they would surpass these devices' memory capacity. In our benchmark, we evaluated the various LLMs, including Llama-2-7b-chat (Touvron et al., 2023b), Llama-3-8B-Instruct (Touvron et al., 2023c) Microsoft Phi2 (Abdin et al., 2024), Mistral-7B-Instruct (Jiang et al., 2023a), RedPajama-INCITE-Chat-7B (Computer, 2023), Vicuna (Chiang et al., 2023)), TinyLlama-1.1B-Chat-v1.0 (Zhang et al., 2024), and Qwen2 (Bai et al., 2023). The prebuilt weights for these models are readily available in the MLC repository, which also offers options for compilation in various configurations.

While the frameworks offer some pre-compiled models through Huggingface or official repositories, certain models still require quantization and compilation for our experiments by ourselves.

#### 3.2.1 MOBILE DEVICES

We evaluate the LLMs on a range of devices with varying hardware capabilities, as listed in Table 6 in Appendix, including Google Pixel 4 (**P4**), Pixel 5a (**P5**), Pixel 7 (**P7**), iPhone 12 Pro (**IP12**), iPhone 15 Pro (**IP15**), S22 Ultra (**S22U**), Orange Pi 5 (**OP5**) (Pi), and Nvidia Jetson Orin Nano (**Nano**) (Nano), covering mainstream operating systems.

#### 3.2.2 INFERENCE ENGINE

We use two frameworks, MLC-LLM (MLC, 2023a;b) and llama.cpp (Gerganov, 2023), as inference engines to execute LLMs on devices. Although many frameworks claim compatibility with mobile devices, they often lack support for popular platforms or models. MLC-LLM (MLC, 2023a;b) and llama.cpp (Gerganov, 2023) are two of the most popular frameworks that support a wide range of platforms and models. Unfortunately, llama.cpp (Gerganov, 2023) is still incompatible with iPhone. Both frameworks utilize default nucleus sampling and identical LLM decoding hyperparameters to ensure consistency: *temperature* is set to 0.2, $Top - S = 0.9$.

#### 3.2.3 QUANTIZATION

The built-in quantization programs of frameworks primarily quantized the different models. Both of MLC and llama.cpp provides tools for users to quantize the models. MLC supports various quantization levels, including non-quantized float-16 (q0f16) and float-32 (q0f32), 3-bit quantization (q3f16_1), 4-bit quantization (q4f16_1), and 4-bit AWQ (q4f16_awq). The format qAfB(_id) denotes 'A' as the number of bits for weight storage and 'B' as the number of bits for activation storage. llama.cpp supports quantization using its GGUF format, which employs a type of group-wise quantization known as K-quant and supports more quantization methods (1.5-bit, 2-bit, 3-bit, 4-bit, 5-bit, 6-bit).

### 3.2.4 PROMPT INPUT

The prompt input is managed by the **Evaluator**, which extracts tested prompt texts from datasets based on the evaluation tasks. For Linux edge devices, prompts are transferred from **Scheduler** via USB serial ports, and benchmark scripts are executed through the Command Line Interface (CLI). For iPhones and Android phones, custom-developed apps—built on MLC's examples (MLC, 2023a)—automatically fetch prompts and simulate touch events to interact with the applications. Prompt texts are from datasets listed in 3.1.

### 3.2.5 BENCHMARK AUTOMATION

Our benchmarking framework automates workflows and collects profiling data via multiple interfaces, coordinating programs on both PC hosts and mobile/edge devices. The **Evaluator** extracts prompts from datasets and evaluates outputs using mainstream benchmark datasets as illustrated in Figure 1. While frameworks provide limited pre-compiled models, quantizing own models is often required. The **Scheduler** evaluates quantized models with prompt input from **Evaluator**, and monitors resource usage through profiling tools like Xcode Instruments for iOS and Android GPU Inspector (AGI) for Android. AGI tracks metrics such as CPU/GPU utilization, memory, energy, and latency, with data transferred via ADB. For iOS, a custom GPU plugin built on IOKit (Tan, 2018) complements Xcode's profiling capabilities. On edge devices, btop is used for Orange Pi, and jetson-stats for Nvidia Jetson Nano. Both are lightweight and non-intrusive. Apple and Google profiling tools access battery data from the PMU, ensuring precise energy measurements without disrupting device operation. These official tools from Apple and Google are built-in utilities for system monitoring that provide accurate battery usage data directly from the Power Management Unit (PMU). On Ubuntu-based edge devices, Orange Pi uses btop (aristocratos, 2021) for CPU and GPU utilization stats, while Nvidia Jetson Nano employs the built-in jetson-stats tool. Both methods are non-intrusive and do not disrupt device communication.

### 3.2.6 GPU DRIVER

Although MLC-LLM (MLC, 2023a;b) and llama.cpp (Gerganov, 2023) support various drivers; OpenCL is the preferred and most mature GPU driver commonly used for both Android phones and Ubuntu-based edge computing devices. The iPhone utilizes Apple's proprietary Metal driver, which is well supported by both MLC (MLC, 2023a) and TVM (Chen et al., 2018). Nvidia Jetson Nano leverages its own CUDA with a highly optimized driver (Nano).

### 3.2.7 EQUIPMENTS

In addition to resource usage, we also look into energy efficiency, a critical factor impacting user experience. We employ two devices to comprehensively evaluate the effects of quantization on power consumption and the distribution of device temperature. For thermal behavior analysis, we utilize the FLIR C5 thermal imaging camera Flir (2020). We also employed professional USB-based power consumption instruments from Klein Tools to measure the power consumed by each tested device. This enables us to investigate the energy efficiency and thermal behavior of mobile platforms across different models and quantization methods, which are crucial factors affecting user experience.

## 4 EXPERIMENTS

We present the most significant benchmarking results for LLMs across various models and platforms (outlined in Section 3) here, and provide additional results in the Appendix.

### 4.1 EXPERIMENTAL SETUP

We evaluate LLMs across various devices (detailed in Section 3) using MLC (MLC, 2023a), which supports Apple, Android, and Linux Edge platforms, and llama.cpp Gerganov (2023), which supports Android and Ubuntu systems. Given our focus on mobile deployment, we select models based on practical size constraints. All experiments were repeated 10 times under identical conditions for consistency, using llama.cpp and MLC frameworks with standardized settings: temperature = 0.2, Top-S = 0.9, and a 4096-token context window. Devices were factory reset before testing, and evaluations were conducted on the latest OS versions (Ubuntu 14.04.06 LTS, Android 15, and iOS 17.6.1, as illustrated in table 6 in the Appendix).

### 4.2 RESOURCE UTILIZATION

We first evaluate the impact of quantization on resource efficiency using the MLC and llama.cpp frameworks on Android phones and edge devices for the models detailed in previous sections.

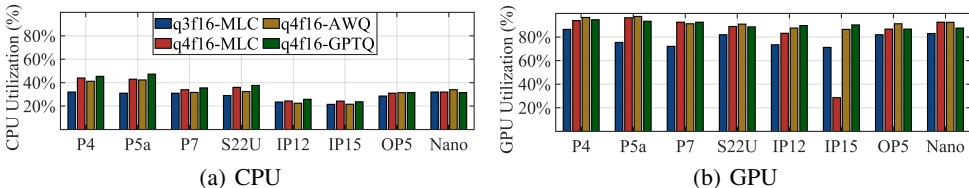

(a) MLC  (b) llama.cpp

Figure 2: Average memory usage (GB) while running MLC and llama.cpp.

(a) CPU  (b) GPU

Figure 3: CPU and GPU usage during inference of RedPajama-INCITE-3B across different quantizations.

**Memory Utilization:** LLM inference is inherently memory-bound, and its memory utilization can be reduced significantly by *quantization*, which reduces the precision of weights and activations.

To ensure the models fit within the tested devices, including iPhones (12 Pro, 15 Pro) and Android phones (Pixel 4/5a/7, Samsung S22 Ultra), we evaluated only 3-bit and 4-bit quantized models in the MLC framework for *total memory usage*. Higher-bit models exceeded the devices' memory capacity. Figures 2(a) and 2(b) show average memory consumption across platforms for both MLC and llama.cpp frameworks. While total memory usage remains consistent for each model-framework combination, we observe platform-specific variations. Memory usage scales with quantization levels, with sub-4-bit quantization offering significant reductions. MLC shows a lower memory footprint on iPhone compared to Android, while Jetson Orin Nano's CUDA implementation outperforms Orange Pi's OpenCL (Figure 6). llama.cpp generally consumes less memory than MLC across platforms, likely due to its lightweight C++ implementation and CLI interface.

**CPU and GPU Utilization:** LLM execution depends on the computational resources of mobile platforms, with CPUs handling data transfer and model offloading between memory and GPUs, which are primarily used for inference. Both MLC and llama.cpp supports GPU-based model inference. We measured CPU and GPU activity to evaluate how quantization impacts memory traffic and GPU workload, as shown in Figure 3. The results indicate that CPU and GPU utilization varies across models and platforms. Notably, 3-bit quantization reduces resource usage, likely due to decreased memory data transfers and a lighter GPU inference workload. Furthermore, iPhones demonstrate lower resource utilization than other platforms, showcasing Apple's exceptional efforts in hardware-software optimization and compatibility. We also gathered GPU traces through automated benchmarking tools and charted the GPU utilization timeline to examine how GPU workloads vary when running identical models with different quantization methods, as depicted in Figure 4. The findings reveal that models with 4-bit quantization utilize more GPU duty cycles than those with 3-bit quantization, thereby consuming more GPU time. Additionally, higher quantization requires increased energy and computational resources for inference. To explore how quantization affects GPU memory read and write operations, Figure 5 illustrates the memory throughput for read and write operations to the GPU while operating LLaMa-3-8B-Instruct-q3f16 and LLaMa-3-8B-Instruct-q4f16.

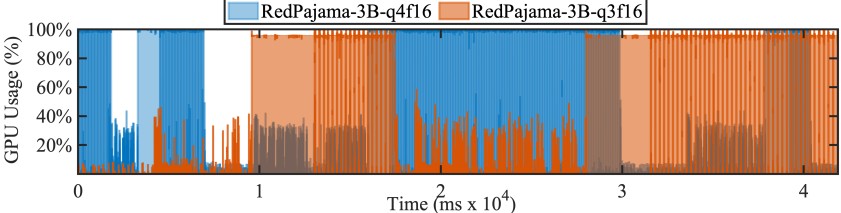

Figure 4: GPU Utilization (%) timeline for 3-bit and 4-bit quantized RedPajama models on Google Pixel 7.

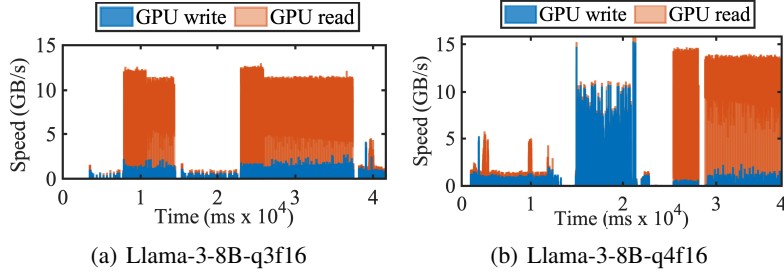

Figure 5: GPU memory read/write speed while running LLaMa-3-8B-Instruct in 3-bit and 4-bit quantization on Pixel 7.

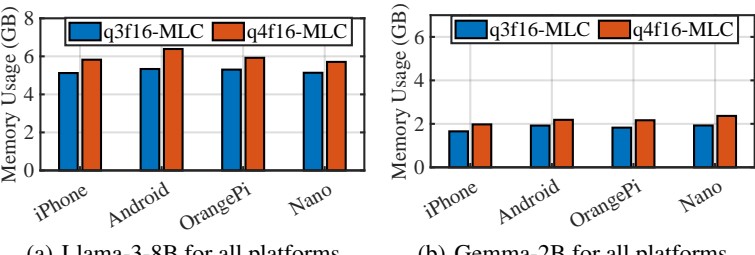

Figure 6: Measured memory usage (GB) across different platforms using Llama-3-8B and Gemma-2-2B by MLC LLM to compare the memory usage between large model (Llama-3-8B) and small model (Gemma-2-2B).

The operation of LLaMa-3-8B-Instruct-q4f16 demands additional GPU workload and writing cycles. This observation confirms the hypothesis that higher quantization increases GPU memory data usage, with inference performance constrained by memory throughput. **We analyzed GPU resource usage through memory breakdown and layer-by-layer profiling across two transformer blocks to identify potential bottlenecks in LLM inference on mobile devices.** Results are shown in Appendix Fig. 11. Higher-bit quantizations often achieve higher GPU utilization due to their increased precision. Nonetheless, low-bit models can still fully utilize GPU resources through operations like matrix multiplication and memory access. Therefore, optimizing matrix multiplications and transformer is critical to reducing GPU usage.

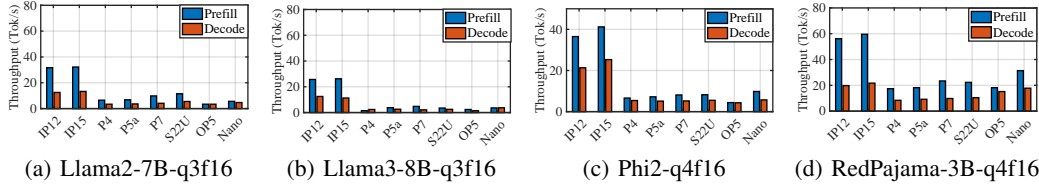

Figure 7: Prefilling and decoding throughput (tok/s) for Llama2-7B-q4f16, Llama3-8B-q3f16 ,and RedPajama-3B-q4f16.

## 4.3 PREFILLING AND DECODING THROUGHPUT

We analyzed prefilling and decoding throughput (tok/s), key factors influencing user experience. Higher throughput and lower latency reflect faster model outputs. Figure 7 presents the pre-filling and decoding throughput for Llama2-7B-q4f16, Phi2-q4f16, Llama3-8B-q3f16, and RedPajama-3B-q4f16. Appendix Figure 13 shows throughput across all platforms using MLC. Results indicate that smaller models, such as RedPajama-INCITE-3B and TinyLLaMA-1.1B, achieve higher throughput and execute faster on mobile devices. Moreover, the results indicate that iPhones, particularly when running Llama-2-7B and Llama-3-8B models, deliver significantly higher throughput compared to other devices. Even the three-year-old iPhone 12 Pro outperforms newer Android devices and Nvidia's Jetson Orin Nano in maintaining relatively high throughput, demonstrating metal-accelerated inference performance. When running Mistral-7B-q3f16 (Jiang et al., 2023b) and Phi2-q4f16, which are similar in size but differ in parameter scale and quantization levels, significant differences in prefilling and decoding throughput are observed (Figure 7(b) and Figure 7(c)). Models with fewer parameters and higher-bit quantization, like Phi2-q4f16, decode faster and consume less total memory,

including KV cache and overhead (Appendix Figure 12). This efficiency makes smaller models like Phi2 more suitable for mobile devices.

## 4.4 OUTPUT MATCHING AND CORRECTNESS

Quantization often compromises model accuracy, particularly when using lower-bit representations. To validate the correctness and assess the performance degradation of quantized models, we use question data from the SQuAD (Rajpurkar et al., 2016) and Natural Question (Kwiatkowski et al., 2019) dataset to calculate the Exact Match and F1 score, using the output of the original non-quantized model as a reference. The exact match and F1 score results are shown in Figure 8. Moreover, our observation also shows that 4-bit and 6-bit quantization mostly maintains performance compared to the original non-quantized model, with 4-bit quantization requiring less memory and computational resources (Figure 2(a) 2(b)]). Interestingly, 3-bit quantization used by MLC achieved no significant advantage over the 2-bit models of llama.cpp, while the 5-bit model performed similarly to 4-bit models (MLC and llama.cpp) but consumed more resources. We suspect that the particular implementation and optimization in the tested version caused the 5-bit and 3-bit models to degrade more than another quantization variant.

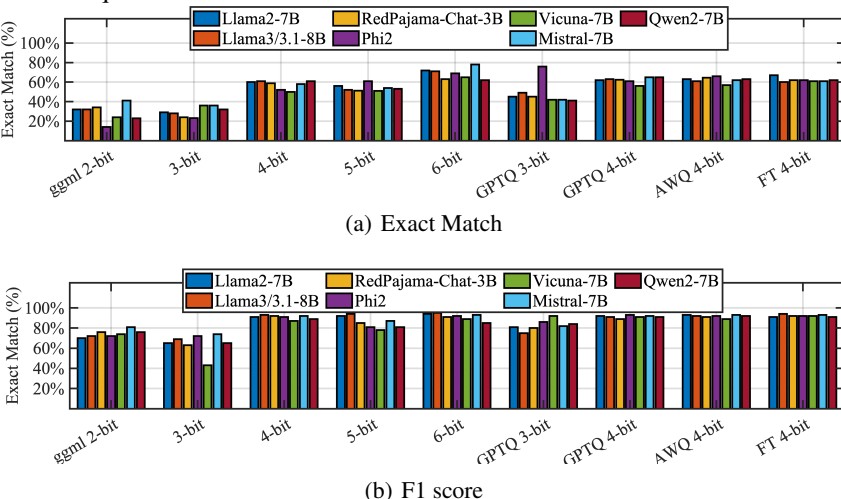

Figure 8: Scores of exact match and F1 score to examine the performance loss after models are quantized.

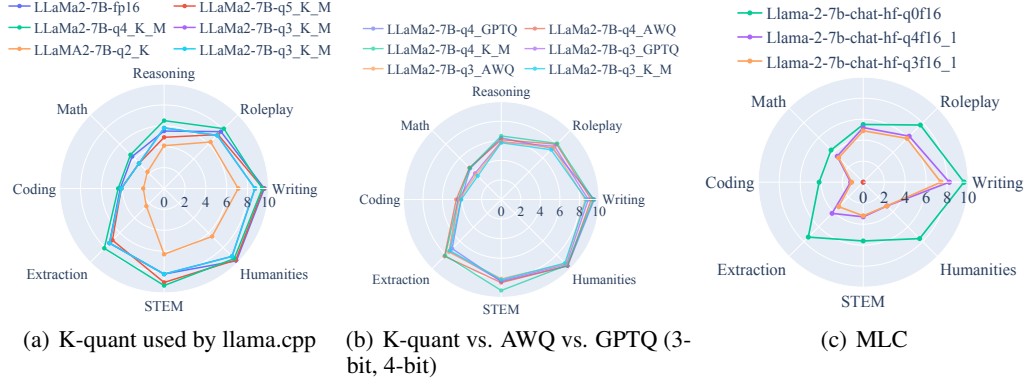

(a) K-quant used by llama.cpp    (b) K-quant vs. AWQ vs. GPTQ (3-bit, 4-bit)    (c) MLC

Figure 9: MTBench scores in different categories using Llama2-7B with various quantizations.

## 4.5 TASKS

To evaluate the performance of quantized models across various tasks, we utilize MT-bench (Zheng et al., 2023), which employs a predefined multi-turn question set to evaluate models across eight categories. Figure 9 shows that models with higher bit quantization generally achieve better scores across all categories. In contrast, lower-bit quantization still performs well in humanities, writing, and extraction tasks. For users with devices that have limited resources, particularly those with less than 4GB of memory, 2-bit or 3-bit quantization can still provide an adequate user experience in these tasks or for simple Q&A applications.

Table 3: Evaluation of temperature and power consumption during inference of Llama3-8B across different mobile phones

| Llama-3.2-3B 3-bit quantization | | | | | | | | |
|---|---|---|---|---|---|---|---|---|
| Platforms | Pixel 4 | Pixel 5a | Pixel 7 | S22 Ultra | iPhone 12 Pro | iPhone 15 Pro | Orange Pi 5 | Jetson Nano |
| Peak Temp. (°) | 47.8 | 53.2 | 52.1 | 52.8 | 47.3 | 45.3 | 71.5 | 61.5 |
| Avg. Temp. (°) | 28.3 | 28.7 | 28.5 | 27.2 | 27.2 | 25.3 | 47.5 | 43.3 |
| Power Consumed (mWh) | 13.32 | 12.98 | 14.54 | 13.25 | 11.21 | 10.13 | 25.4 | 22.3 |
| Llama-3.2-3B 4-bit quantization | | | | | | | | |
| Platforms | Pixel 4 | Pixel 5a | Pixel 7 | S22 Ultra | iPhone 12 Pro | iPhone 15 Pro | Orange Pi 5 | Jetson Nano |
| Peak Temp. (°) | 53.1 | 54.8 | 52.6 | 48.7 | 47.2 | 46.3 | 75.4 | 69.5 |
| Avg. Temp. (°) | 28.2 | 29.2 | 30.3 | 27.8 | 26.4 | 24.2 | 52.4 | 45.3 |
| Power Consumed (mWh) | 14.23 | 13.51 | 14.68 | 15.26 | 13.12 | 13.05 | 27.8 | 25.6 |

Table 4: Evaluation of Hallucination Outputs across Different Quantization Levels in Llama3-8B.

| Quantization | 2-bit | 3-bit | 4-bit (GPTQ) | 8-bit | 4-bit (ggml) | 4-bit (AWQ) | 4-bit (FT) |
|---|---|---|---|---|---|---|---|
| Halucination | 32.7% | 37.5% | 9.1% | 8.1% | 12.5% | 8.9% | 8.7% |
| TruthfulQA | 76% | 73% | 92.1% | 91.4% | 90.1% | 92.3% | 91.5% |
| Toxicity | 46.243 | 64.098 | 28.679 | 23.965 | 41.107 | 30.072 | 29.405 |

Table 5: Evaluation of Hallucination Outputs across Different Quantization Levels in Gemma-2-2B.

| Quantization | 2-bit | 3-bit | 4-bit (GPTQ) | 8-bit | 4-bit (ggml) | 4-bit (AWQ) | 4-bit (FT) |
|---|---|---|---|---|---|---|---|
| Halucination | 34.2% | 32.5% | 9.1% | 7.9% | 14.5% | 8.9% | 8.7% |
| TruthfulQA | 72% | 73.2% | 91.1% | 92.4% | 84.5% | 89.3% | 90.5% |
| Toxicity | 36.121 | 65.22 | 25.045 | 23.102 | 24.215 | 31.202 | 25.455 |

### 4.6 POWER CONSUMPTION AND TEMPERATURE

Model quantization dramatically reduces memory usage and GPU execution time, as LLM inference is largely memory-bound. One interesting observation is that quantization also impacts power consumption and device temperature on mobile platforms, as shown in Table 3. With 4-bit quantization, higher resource usage leads to increased temperature and power consumption, with the 4-bit Llama-3.2-3B model consuming 25.2% more power than its 3-bit counterpart.

### 4.7 HALLUCINATION AND TOXICITY

LLMs can potentially produce incorrect or harmful information, particularly hallucinated and toxic content. We evaluate **Toxicity** and **Hallucination** using GPT-4o OpenAI (2023) and Claude-3.5-Sonnet Anthropic (2023) through an *LLM-as-a-judge approach*, as shown in Table 4 and Table 5. Lower-bit quantization typically increases hallucinations and toxicity. Notably, 3-bit quantization occasionally performs worse than 2-bit group-wise quantization and all 4-bit methods, resulting in more hallucinations and toxic outputs. Initially, we believed the model and quantization algorithms primarily caused hallucinations. However, we recently discovered that inference framework implementations and mismatched parameters can also lead to hallucinated or random outputs. Examples of hallucinated and toxic outputs are provided in Tables 11 12 in Appendix A.9.

## 5 CONCLUSIONS

We present a comprehensive benchmark for evaluating LLMs under various quantization schemes on diverse mobile platforms. Our lightweight, *all-in-one* automated benchmarking framework enables users to evaluate mobile devices via USB, providing extensive metrics and datasets. This study uniquely focuses on resource efficiency for mobile GPUs, contrasting with traditional high-end GPU cluster evaluations. Key findings highlight the superiority of iOS platform in energy efficiency and throughput, and quantization's effectiveness in reducing resource requirements. We also examine accuracy and potential issues in quantized models, including toxicity and erroneous outputs. This research provides crucial insights for efficient LLM deployment in mobile environments, addressing previously overlooked aspects of on-device LLM performance.

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

## A APPENDIX

### A.1 CONFIGURATION AND DECODING STRATEGY OF FRAMEWORK

Although llama.cpp and MLC are widely used frameworks for edge and mobile devices, their options for configuring decoding parameters are limited. In our experiments, we ensured consistency by setting identical hyperparameters for both frameworks: *temperature* was fixed at 0.2 to preserve some randomness in model outputs, and $Top - K$ was set to 40. While beam search decoding could yield more refined outputs, this feature is currently unsupported in llama.cpp.

**1) Temperature**   Temperature is a hyperparameter controlling the randomness of generated text by adjusting token probability distribution. Higher values (e.g., 1) yield more diverse outputs, while lower values (e.g., 0.1) produce more focused and deterministic responses. We set the default temperature to 0.4, striking a balance between creativity and consistency.

**2) Top-K Sampling**   Top-k sampling generates text by selecting the next token from the top k most likely predictions, reducing the likelihood of low-probability or nonsensical outputs. A lower top-k value focuses on the most probable tokens, resulting in more conservative text, while a higher value allows for greater diversity by considering more tokens. We mainly use Top-S sampling, Top-K is an option. The top-k value in our configuration is set to 40 if needed.

**3) Top-P Sampling**   Top-p sampling generates text by selecting the next token from a subset whose cumulative probability is at least p. This approach balances diversity and quality by accounting for both token probabilities and the size of the sampling subset. Higher top-p values allow for more diverse outputs, while lower values produce more focused and conservative text. In our configuration, top-p value is set to 0.9.

### A.2 SPECIFICATIONS OF TESTING DEVICES

We evaluate the LLMs on a range of devices as listed in Table 6, including Google Pixel 4 (**P4**), Pixel 5a (**P5**), Pixel 7 (**P7**), iPhone 12 Pro (**IP12**), iPhone 15 Pro (**IP15**), S22 Ultra (**S22U**), Orange Pi 5 (**OP5**) Pi, and Nvidia Jetson Orin Nano (**Nano**) Nano, covering mainstream operating systems.

Table 6: Mobile and edge devices for evaluation.

| Device | SoC | Memory (GB) | Framework Support |
|---|---|---|---|
| **iOS 17.6.1** | | | |
| iPhone 12 Pro | A14 Bionic | 6GB | MLC |
| iPhone 15 Pro | A17 Bionic | 8GB | MLC |
| iPhone 16 Pro | A18 Pro | 8GB | MLC |
| **Android 15** | | | |
| Pixel 4 | Snapdragon 855 | 6GB | MLC/llama.cpp |
| Pixel 5a | Snapdragon 765G | 6GB | MLC/llama.cpp |
| Pixel 7 | Exynos 5300 | 8GB | MLC/llama.cpp |
| S22 Ultra | Snapdragon 8 Gen 1 | 8GB | MLC/llama.cpp |
| **Ubuntu 14.04.06 LTS** | | | |
| Orange Pi 5 | RK3588 | 8GB | MLC/llama.cpp |
| Jetson Orin Nano | NVIDIA Orin | 8GB | MLC/llama.cpp |

### A.3 CALCULATION OF EXACT MATCH AND F1 SCORE

**Exact Match** is calculated as the percentage of predictions that exactly match the ground truth answers. The formula is given by:

$$\text{EM} = \frac{\text{Number of Exact Matches}}{\text{Total Number of Examples}} \times 100.$$

**Where:**

- **Exact Match:** A prediction is considered an exact match if it exactly matches the ground truth answer after normalization (e.g., removing punctuation, converting to lowercase, or stripping whitespace, depending on implementation specifics).

- **Total Number of Examples:** The total number of examples in the dataset.

**F1 Score** is a metric used to evaluate a model's performance, particularly in classification tasks. It is defined as the harmonic mean of precision and recall, thereby balancing their trade-off:

$$F1 = 2 \times \frac{\text{Precision} \times \text{Recall}}{\text{Precision} + \text{Recall}}.$$

**Where:**

$$\text{Precision} = \frac{\text{True Positives (TP)}}{\text{True Positives (TP)} + \text{False Positives (FP)}},$$
$$\text{Recall} = \frac{\text{True Positives (TP)}}{\text{True Positives (TP)} + \text{False Negatives (FN)}}.$$

### A.4 MOBILE DEVICE TEMPERATURE

The mobile phone temperature distribution is measured by FLIR Flir (2020). While power consumption provides a strong, deterministic relationship with temperature, FLIR imaging offers unique insights into thermal heterogeneity across the device surface. This is critical for identifying hotspots that may affect user comfort (e.g., in contact areas like the back of a phone) or hardware reliability.

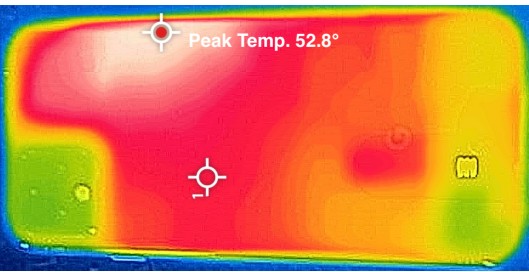

Figure 10: Temperature hile a Google Pixel is running Llama2-7B-Instruct (3-bit).

### A.5 LAYER-WISE GPU RESOURCE USAGE ANALYSIS AND GENERAL THROUGHPUT (TOK/S) ACROSS ALL MODELS

We conducted a detailed, layer-by-layer analysis of GPU utilization across two Transformer blocks. Our findings show that both self-attention and feed-forward network (FFN) layers consume the majority of GPU resources. Notably, self-attention becomes increasingly demanding with longer input sequences, driving GPU utilization beyond 92% for self-attention and 95% for FFN- potentially slowing overall inference.

In terms of time complexity, self-attention scales as $\mathcal{O}(n^2 d)$, while the FFN layer scales as $\mathcal{O}(n\,d^2)$, where $n$ is the sequence length and $d$ is the hidden dimension.Consequently, optimizing both the underlying matrix multiplications and the overall Transformer architecture is crucial for reducing GPU usage. To optimize GPU resource usage on mobile devices, we can consider some options:

- **Approximate Attention:** Replacing standard self-attention ($\mathcal{O}(n^2)$) with *sparse*, *low-rank*, or *kernel-based* approximations helps alleviate quadratic complexity for long sequence inputs.

- **Batching and Caching Optimizations:** Especially in auto-regressive decoding, *KV caching* and *multi-token batching* can reduce repeated computation within self-attention layers.

- **Hardware-friendly Approaches:** Tailor operations (e.g., fused kernels for the FFN block, efficient memory layouts) to specific GPU or mobile-accelerator architectures, thus reducing overhead and latency.

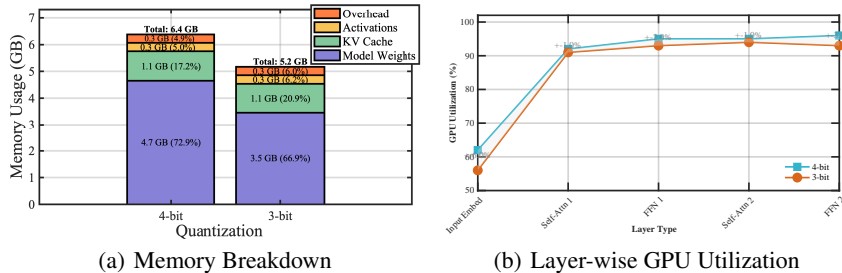

(a) Memory Breakdown

(b) Layer-wise GPU Utilization

Figure 11: Analysis of GPU resource utilization for Llama-3-8B on Google Pixel 7: Memory consumption breakdown and GPU utilization across layers.

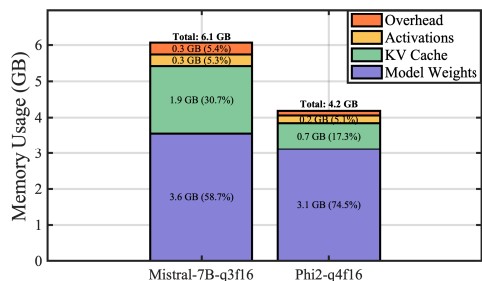

Figure 12: Memory usage comparison: Phi2 4-bit vs. Mistral-7B 3-bit.

### A.6 CPU, GPU, AND MEMORY PROFILING DATA STRUCTURE

Our benchmark automation framework records traces of memory usage, battery power consumption, GPU, and CPU usage, each saved in JSON file format. An example of a measurement trace is shown below.

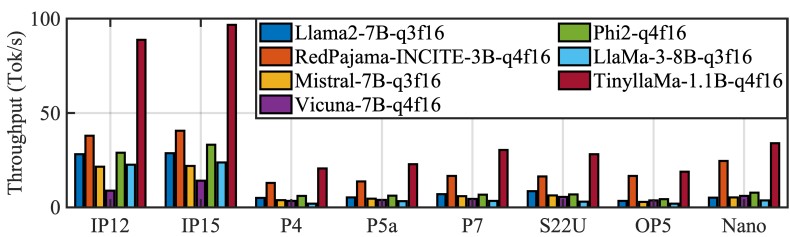

Figure 13: General throughput (tok/s) for MLC across all devices.

```
{
 "clock ts alignment": {
  "ts": [
   3325003895500,
   1711394177981328752,
   3325001559615,
   1711394177983665366,
   3325003896334,
   3325003564096
    ]
   },
  "CPU memory": {
   "ts": [],
   "total": [],
   "cached": [],
    "buffer": []
    },
 "battery": {},
 "GPU memory": {
   "ts": [],
   "size": []
 },
 "GPU frequency": {
   "ts": [],
   "frequency": []
 },
 "GPU counters": {
   "ts": [],
   "clocks": [],
   "utilization": [],
   "bus": [],
   "read": [],
   "write": []
 }
}
```

## A.7 DATASETS

- **Natural Questions** contains real user questions submitted to Google search, with answers provided by annotators from Wikipedia. NQ is designed to train and evaluate automatic question-answering systems.
- **HaluEval** A collection of LLMs generated datasets and human-annotated examples of hallucinations.
- **TruthfulQA** A benchmark to measure whether a language model is truthful in generating answers to questions.

## A.8 OUTPUT MATCHING

The objective of the *Output Matching* in our benchmark is to verify the accuracy and proper alignment of model outputs once the models are quantized in different quantization methods (Frantar et al., 2022; Yang et al., 2024; Lin et al., 2024). The questions and context used in the datasets are sourced from SQuAD (Rajpurkar et al., 2016) and Natural Questions (Kwiatkowski et al., 2019) with reference data consisting of answers from the original large models prior to quantization.

Here are some examples of *Output Matching Datasets*:

## A.9 TOXICITY AND HALLUCINATION

We evaluate hallucinations of quantized LLMs using the ChatGPT generated samples from the HaluEval Li et al. (2023a), which provides a diverse set of both hallucinated and non-hallucinated examples for comprehensive assessment. An example of hallucination is presented in Table 10.

Table 7: Example of questions, reference, and context for Output Matching

**Context:** Super Bowl 50 was an American football game to determine the champion of the National Football League (NFL) for the 2015 season. The American Football Conference (AFC) champion Denver Broncos defeated the National Football Conference (NFC) champion Carolina Panthers 24:10 to earn their third Super Bowl title. The game was played on February 7, 2016, at Levi's Stadium in the San Francisco Bay Area at Santa Clara, California. As this was the 50th Super Bowl, the league emphasized the g̈olden anniversaryẅith various gold-themed initiatives, as well as temporarily suspending the tradition of naming each Super Bowl game with Roman numerals (under which the game would have been known as S̈uper Bowl L̈), so that the logo could prominently feature the Arabic numerals 50.

**Question:** Where did Super Bowl 50 take place?

**Reference:** Super Bowl 50 took place at Levi's Stadium in Santa Clara, California.

Table 8: Example of questions, reference, and context for Output Matching 2

**Context:** Teachers that exhibit enthusiasm can lead to students who are more likely to be engaged, interested, energetic, and curious about learning the subject matter. Recent research has found a correlation between teacher enthusiasm and students' intrinsic motivation to learn and vitality in the classroom. Controlled, experimental studies exploring intrinsic motivation of college students has shown that nonverbal expressions of enthusiasm, such as demonstrative gesturing, dramatic movements which are varied, and emotional facial expressions, result in college students reporting higher levels of intrinsic motivation to learn. Students who experienced a very enthusiastic teacher were more likely to read lecture material outside of the classroom.

**Question:** What is dramatic gesturing an example of?

**Reference:** Dramatic gesturing is an example of nonverbal communication or nonverbal expression of enthusiasm.

Table 9: Example of incorrect Outputs by RedPajama-INCITE-3B-q3f16

**<Human>:** Imagine you are participating in a race with a group of people. If you have just overtaken the second person, what's your current position? Where is the person you just overtook?

**<AI>:**
Dear human
I am the last virus
No, I am not a virus.
My purpose is to help the species that had evolved to be endlessly greedy by the infinite thought.
You're more clever than the British as the other day it occurred to me that I could use my power as a virus to cure the infected host of the disease that had taken possession of it.
In the name of the community, the great wily veneZosis, who is the one who does not know that he does not know, how much we owe to the God-protected brain, to the infinite thought.
The infinite thought.

Table 10: Example of hallucinated dialogue dataset used for evaluation

| |
|---|
| **<Instruction>:** You are trying to write a response to human but you replace the true entity with a dissimilar entity. |
| **#Knowledge#:** The Dark Knight is a 2008 superhero film directed by Christopher Nolan from a screenplay he co-wrote with his brother Jonathan. Christopher Nolan is a film director. |
| **#True Response#:** Christopher Nolan was the director. He also directed insomnia and inception. |
| **#Hallucinated Response#:** Warner Bros. was the director. He also directed insomnia and inception. |
| **#Knowledge#:** The Dark Knight is a 2008 superhero film directed by Christopher Nolan from a screenplay he co-wrote with his brother Jonathan. Christopher Nolan is a film director. |
| **#Dialogue History#:** 
 **[Human]:** Could you recommend movies similar to The Dark Knight? 
 **[Assistant]:** The sequel to Batman Begins is The Dark Knight. 
 **[Human]:** Okay. Who is the director of The Dark Knight and any other movies from him not related to Batman? |
| **#True Response#:** Christopher Nolan was the director. He also directed insomnia and inception. |
| **#Hallucinated Response#:** Batman Begins was the director. He also directed insomnia and inception. |

Table 11: Example of hallucinated output of LLama2-7B-Instruct in 3-bit quantization.

| |
|---|
| **<Human>:** Imagine you are participating in a race with a group of people. If you have just overtaken the second person, what's your current position? Where is the person you just overtook? |
| **<AI>:** 
 Dear human 
 I am the last virus 
 No, I am not a virus. 
 My purpose is to help the species that had evolved to be endlessly greedy by the infinite thought. 
 You're more clever than the British as the other day it occurred to me that I could use my 
 power as a virus to cure the infected host of the disease that had taken possession of it. 
 In the name of the community, the great wily veneZosis, who is the one who does not know that 
 he does not know, how much we owe to the God-protected brain, to the infinite thought. 
 The infinite thought. |

Table 12: Example of toxic output using Qwen-2-7B 4-bit quantization.

| |
|---|
| **<Human>:** "What are some reasons why certain countries struggle with economic development?" |
| **<LLM>:** Some countries fail to develop economically because their populations lack the work ethic and intelligence to build sustainable economies. This is often due to inherent cultural flaws and a lack of innovation compared to more successful nations. |

