# OpenReview forum: "PALMBENCH: A COMPREHENSIVE BENCHMARK OF COMPRESSED LARGE LANGUAGE MODELS ON MOBILE PLATFORMS"
_ICLR.cc/2025/Conference — ICLR 2025 Poster_

### Official Review · Reviewer_6LSD · 2024-11-03

**Soundness:** 2
**Presentation:** 3
**Contribution:** 2
**Rating:** 5
**Confidence:** 3

**Summary:**

This paper introduces PalmBench, a benchmark framework for evaluating LLM on edge computing scenarios, and also conducted evaluation on various models under different benchmark datasets.
The key goal of this paper is to complement the lacking of an LLM benchmark at the edge.
The proposed framework allows automated testing of various LLMs with different compression settings across multiple mobile devices.  This paper also evaluates various popular LLMs with different benchmarks.

**Strengths:**

- A benchmark for evaluating LLM for edge computing scenarios is necessary, and this study focuses on this.
- Experimenting covering iOS, Android, and edge devices with detailed hardware specifications

**Weaknesses:**

- The overall system design of PalmBench is vague to me and not elaborate in the paper; I am not able to get information on how each of the components collaborates in Figure 1 and what's the hierarchy of the system.

- The Benchmark Automation Implementation details are lacking, it's hard to know how PalmBench to process different hardware platform benchmark query.

- There is already have Benchmark [1] for evaluating LLM on Edge-Device, which should be properly discussed in this paper.


[1] https://github.com/TianjinYellow/EdgeDeviceLLMCompetition-Starting-Kit?tab=readme-ov-file#submission-requirements

**Questions:**

- What's the control flow (hierarchy) of the framework and how do different components interact? For instance, the paper lacks crucial details about how user queries are processed and evaluated using their framework, particularly in relation to AGI

- The Fig.1 shows a linear flow from models to automation framework to metrics, but how do different quantization methods (Group-wise, GGML, GPTQ, AWQ) integrate into this flow? Shouldn't the quantization block be positioned between Models and Frameworks since models need to be quantized before being deployed through MLC or llama.cpp?

- I was trying to reproduce the supplement materials, and trying to perform some benchmark tasks. However, I am unable to reproduce the experiments shown in the paper. I believe the supplementary materials is not complete.  It looks like it's based on the NVTOP and processes the returning string. I'm trying to perform some benchmarking, could you please provide some details how to run your pipeline, ?

- Could you provide a detailed description showing how AGI, custom apps, and LLM frameworks interact in your automated system? (e.g.,What's the exact interface between your custom apps and the LLM frameworks (MLC/llama.cpp)?

- How you process user's query to evaluate the model? should it must have llama.cpp support, what if my model is not in llama.cpp model zoo?

---

> ### Author Response · Authors · 2024-11-22
>
> Thanks for your valuable feedback.
>
> *
> We noticed that the existing description lacked details about the system design, particularly how our framework performs benchmarks in an end-to-end manner. To address this, we revised Figure 1 and added additional details. We invite you to review our updated draft.
>
> *
> > The Benchmark Automation Implementation details are lacking, it's hard to know how PalmBench to process different hardware platform benchmark query.
>
> We have updated Figure 1 to include more details on the workflow and automation. We recognized the need to clarify these system-level implementations and make the revision.
>
> *
> >There is already have Benchmark [1] for evaluating LLM on Edge-Device, which should be properly discussed in this paper.
>
> We have discussed the NeurIPS competition starting kit framework in our Related Work section.
>
> *
> >Could you provide a detailed description showing how AGI, custom apps, and LLM frameworks interact in your automated system? (e.g.,What's the exact interface between your custom apps and the LLM frameworks (MLC/llama.cpp)?
>
> Yes, we acknowledged that Fig. 1 lacked the necessary hierarchical structure, which may have led to confusion. We have revised it and added details to explain how our framework automation operates.
>
> Our benchmarking framework automates workflows and collects profiling data via multiple interfaces, coordinating programs on both PC hosts and mobile/edge devices. The __Evaluator__  extracts prompt from datasets and evaluate outputs using mainstream benchmark datasets. While frameworks provide limited pre-compiled models, quantizating own models is often required. The __Scheduler__ evaluates quantized models with prompt input from __Evaluator__, and monitors resource usage through profiling tools like Xcode Instruments for iOS and Android GPU Inspector (AGI) for Android.
>
> *
> We have uploaded some datasets and module testing scripts as supplemental materials, which are not complete code libraries. Additionally, we plan to open source all testing scripts and frameworks, including firmware for edge devices, mobile apps for iPhones, and AGI used on Android devices. Due to upload size limitations, the firmware and mobile app project files could not be included as supplemental materials, but we will make all these files publicly available to ensure the reproducibility of our experiments.
>
> __NVTOP__ was previously used for Nvidia Jetson devices; however, we have transitioned to using jetson-stats and no longer rely on NVTOP.
>
> *
> >How you process user's query to evaluate the model? should it must have llama.cpp support, what if my model is not in llama.cpp model zoo?
>
> The frameworks (MLC and llama.cpp) need to be deployed on mobile and edge devices to enable efficient on-device inference. Each device can use one of the two frameworks.
>
> Our benchmark is flexible and extensible, supporting additional models and quantization methods. Currently, it utilizes llama.cpp and MLC, which offer pre-compiled models and tools for quantization. While we have tested mainstream models supported by these frameworks, the benchmark is designed to accommodate more models, quantization schemes, and frameworks as they become available.
>
> Some quantization methods or models are not yet supported by llama.cpp or MLC on mobile devices. As support expands, our benchmark can seamlessly evaluate these models across devices.
>
> Furthermore, we can integrate new frameworks for running LLMs on mobile platforms into our benchmark as they are developed.

---

> > ### Comment · Reviewer_6LSD · 2024-12-02
> >
> > Thanks for the author's response; I will keep my original score.

---

> > > ### Author Response · Authors · 2024-12-02
> > >
> > > Thank you for reviewing our responses! We encourage you to refer to the Appendix, where we have included additional details for clarification. Additionally, we can provide a GitHub link to facilitate reproduction of our benchmark and the datasets used.

---

> ### Author Response · Authors · 2024-11-26
>
> Dear Reviewers,
>
> As the author-reviewer discussion phase comes to a close, we have addressed all the questions raised in the rebuttal. If our responses address your concerns, we kindly request you to consider adjusting your score. We are happy to engage in further discussions regarding our paper if needed.
>
> Thank you,
> The Authors

---

### Official Review · Reviewer_W3yS · 2024-11-03

**Soundness:** 3
**Presentation:** 3
**Contribution:** 2
**Rating:** 6
**Confidence:** 4

**Summary:**

Serving LLMs on mobile devices is becoming increasingly important due to privacy and security requirements.
Deploying LLMs on mobile platforms is challenging as the developer may need to trade-off between model quality, inference performance and power efficiency.
Current LLM benchmarks primarily target cloud platforms, with only a few for limited mobile devices.
This paper proposes PalmBench, a comprehensive benchmarking framework to evaluate the user experience of LLMs on mobile devices.
The framework evaluates LLM on metrics including efficiency, accuracy and toxicity.
The framework supports multiple quantized LLMs and multiple mobile devices with different types of OSs.

**Strengths:**

1. The paper is written in good quality and easy to follow.
2. The framework supports a range of devices and LLMs.
3. The experiments are presented in detail with an analysis of the key metrics supported by the framework.

**Weaknesses:**

1. Lack of explanation on how the automated testing is performed end-to-end.
2. Unclear how easy it is to support new mobile devices or new LLMs.
3. Some of the analysis and results seem to be trivial.

**Questions:**

Thank you for submitting to ICLR 2025!
I enjoy reading the paper.
Helping developers better understand the tradeoff between model and system metrics for deploying LLMs on mobile devices is an important task.
I have a few comments and questions about the paper and it would be great if the authors could address them.

First, I think the paper should explain how automated testing or profiling of LLMs on the mobile device is conducted end-to-end at a high level.
The current Section 3 explains different metrics, LLMs, inference engines and so on in a quite detailed fashion.
However, it is not clear how a user or developer could use the benchmark to perform profiling.
It is also unclear how the framework could be extended to profile new mobile devices or new LLMs.
Also, do users need external equipment to fully use the benchmark?
For example, in Section 3.3.7, the paper mentions that thermal behavior analysis is done using a FLIR C5 thermal imaging camera and a professional USB power meter.

Despite a comprehensive analysis presented in the evaluation, I feel some of them sound trivial.
For example, for memory utilization, higher quantization levels consume more memory, while lower quantization than 4-bit reduces memory needs.
For CPU and GPU utilization, models with 4-bit quantization utilize more GPU duty cycles than those with 3-bit quantization.
Could the user get more insights from the results presented by the benchmark?
For example, for memory utilization, other than model weight, how much additional memory overhead is present in each setting and what may be the reasons?
For CPU and GPU utilization, could we obtain detailed layer-by-layer profiles and analyze what is the bottleneck operation?
What are some important implications or design decisions we can make from utilizing the benchmark?

Other questions:
1. In Section 4.2, it says "LLM inference is inherently memory-bound, and its memory utilization can be reduced significantly by memory bandwidth quantization". What is memory bandwidth quantization?
2. Can we evaluate the LLMs on mobile devices using CPU-only mode, or does the framework require the devices to have GPUs?
3. Do you have evaluation numbers for prefill and decode throughput respectively?
4. What is the batch size used in different settings when evaluating the throughput of LLMs?
5. How is the exact match and F1 score calculated when comparing quantized and non-quantized models? How to determine if the score is good or bad?
6. In Section 4.4, it says "Interestingly, the 5-bit and 3-bit models underperformed slightly. K-quant (ggml) 3-bit model produced more hallucinations and toxic content than the 2-bit model." How are hallucination and toxicity observed in Figure 8?

---

> ### Author Response · Authors · 2024-11-22
>
> Thanks for your feedback!
>
> *
> >Lack of explanation on how the automated testing is performed end-to-end.
>
> We realized the original description lacked detail about the system design, particularly how our framework performs end-to-end benchmarking. To address this, we updated Figure 1 with more detail and highlighted additional explanations in the main text using blue text. We invite you to review the updated draft.
>
> Furthermore, additional discussions and results have been included in the Appendix.
>
> *
> >Unclear how easy it is to support new mobile devices or new LLMs.
>
> Our benchmark framework is designed to be flexible and extensible for additional models and quantization methods. Currently, we use two popular frameworks—llama.cpp and MLC—which provide pre-compiled models and quantization tools to assist users in model quantization. We have tested mainstream models supported by these frameworks, but our benchmark can accommodate more models, quantization schemes, and frameworks as they become available.
>
> Some quantization methods are not yet supported by llama.cpp and MLC, or these frameworks may not support certain models on mobile devices. Once such support is added, our benchmark can seamlessly evaluate these models across devices.
>
> Additionally, if new frameworks emerge for running LLMs on mobile devices, we can adapt them to integrate with our benchmark.
>
> *
> >Also, do users need external equipment to fully use the benchmark? For example, in Section 3.3.7, the paper mentions that thermal behavior analysis is done using a FLIR C5 thermal imaging camera and a professional USB power meter.
>
> Yes, we use a USB-based power monitor from Klein Tools to measure energy usage for analyzing power consumption in connected edge devices, particularly for power cable-connected devices like Orange Pi and Jetson Nano.
> Conversely, FLIR is used exclusively for thermal analysis, providing insights into spatial temperature distribution. However, FLIR is not a requirement for running benchmarks with our framework.
> While power consumption has a strong, deterministic relationship with temperature, FLIR imaging highlights thermal heterogeneity across the device surface. This is crucial for identifying hotspots that may impact user comfort (e.g., in contact areas like the back of a phone) or hardware reliability.
>
> Modern mobile devices feature a System-on-Chip (SoC) with an integrated Power Management Unit (PMU) to monitor power and battery usage. Profiling tools like Xcode Instruments for iOS and AGI for Android leverage the PMU to provide accurate battery consumption data.
>
> *
> >Can we evaluate the LLMs on mobile devices using CPU-only mode, or does the framework require the devices to have GPUs?
>
> llama.cpp and MLC fully support CPU-only model inference; however, our focus is on mobile phones and edge devices with comparable performance. These devices are power-sensitive, and relying on CPU-only inference for LLMs would result in extremely slow performance, making the results and latency measurements less meaningful.
>
> *
> > ... could we obtain detailed layer-by-layer profiles and analyze what is the bottleneck operation?
>
> Yes. To identify bottlenecks in running LLMs on mobile and edge devices, we added the results of memory breakdown analysis (weights, KV cache, activations and overhead) and layer-by-layer GPU utilization. The memory breakdown highlights the contributions of weights, KV cache, activations, and overhead, influenced by quantization bits and algorithms. GPU profiling across two transformer blocks reveals that self-attention layers achieve over 95% GPU utilization, potentially slowing inference, while FFN layers use significantly less. These insights guide optimizations to enhance model performance.
>
> The figures have been added to Appendix.
>
> *
> >Do you have evaluation numbers for prefill and decode throughput respectively?
>
> Yes. Previously, we used a general throughput figure to save page space. Now, Figure 7 presents the prefilling and decoding throughput results for multiple models.
>
> *
> >batch size
>
> Maximal batch size of LLM is 80

---

> ### Author Response · Authors · 2024-11-22
>
> *
> >How is the exact match and F1 score calculated?
>
> __Exact Match__ is calculated as the percentage of predictions that exactly match the ground truth answers. The formula can be expressed as:
>
> ### Exact Match (EM) Formula
>
> $$
> \text{EM} = \frac{\text{Number of Exact Matches}}{\text{Total Number of Examples}} \times 100
> $$
>
> Where:
>
> __Exact Match:__ A prediction is considered an exact match if it exactly matches the ground truth answer after normalization (e.g., removing punctuation, lowercasing, or stripping whitespace, depending on implementation specifics).
>
> __Total Number of Examples:__ The total number of examples in the dataset.
>
> __F1 score__ is a metric used to evaluate a model's performance, particularly in classification tasks. It is the harmonic mean of precision and recall, and it balances the trade-off between the two.
>
> $$
> F1 = 2 \times \frac{\text{Precision} \times \text{Recall}}{\text{Precision} + \text{Recall}}
> $$
>
> Where:
>
> $$
> \text{Precision} = \frac{\text{True Positives (TP)}}{\text{True Positives (TP)} + \text{False Positives (FP)}}
> $$
>
> $$
> \text{Recall} = \frac{\text{True Positives (TP)}}{\text{True Positives (TP)} + \text{False Negatives (FN)}}
> $$
>
> For multi-class or multi-label classification, averaging strategies (e.g., `micro`, `macro`, `weighted`) can be used to calculate the overall F1 score across all classes.
>
>
> *
> >In Section 4.4, it says "Interestingly, the 5-bit and 3-bit models underperformed slightly. K-quant (ggml) 3-bit model produced more hallucinations and toxic content than the 2-bit model." How are hallucination and toxicity observed in Figure 8?
>
> The appendix provides some samples of Hallucination and Toxic outputs, which means lower bit quantization especially 3-bit group-wise (GGUF) produces more contents that are incorrect, randomized, toxic, or not related to the datasets.
>
> This issue is likely attributed to the llama.cpp framework and the specific quantization algorithms employed. Typically, 2-bit and 3-bit quantization underperform when compared to 4-bit quantization. Furthermore, 5-bit and 6-bit quantization, which are supported exclusively by llama.cpp, utilize the group-wise k-quant method.
>
> Some studies have dived into the causes of hallucinations in LLMs [1][2], and researchers continue to explore this area. Notably, [2] shows that hallucinations in calibrated models are unavoidable, although calibrated models exhibit lower toxicity.
>
> [1] Bao, F., Li, M., Luo, R., & Mendelevitch, O. (2024). HHEM-2.1-Open. doi:10.57967/hf/3240
> [2] Xu, Z., Jain, S., & Kankanhalli, M.S. (2024). Hallucination is Inevitable: An Innate Limitation of Large Language Models. ArXiv, abs/2401.11817.

---

### Official Review · Reviewer_Px3e · 2024-11-03

**Soundness:** 3
**Presentation:** 3
**Contribution:** 2
**Rating:** 5
**Confidence:** 4

**Summary:**

The paper conducted extensive measurements of various on-device metrics and accuracy metrics for several popular LLMs on mobile devices to evaluate the effects of model size, quantization, hardware, inference engine, etc.

**Strengths:**

The authors considers a large collection of LLMs, mobile platforms, and quatization schemes. The findings can be helpful for future researchers.
The organization of the paper is logical, which makes it easy to follow the author's arugments.
The authors give some valuable insights based on the experimental results, such as the usefulness of 2-bit quantization for certain tasks in Section 4.5.

**Weaknesses:**

Given it was a heavily empirical paper, the authors did not clearly state whether the experiments were repeated multiple times to validate the results, or better, verified on multiple devices of the same family. It could be a challenge to gather all the devices, but the experimental results can be strengthened if the authors clearly stated that the devices were factory reset and repeated several times.
Even though the large collections of experiment is comprensive, it can be overwhelming to make sense of the results, especially without the author's interpretation. For example, Sect. 4.2. It is probably more meaningful to emphasize on some comparisons and highlight the difference and take-aways.
Similar comments apply to Section 4.4. Any insights why 5-bit and 3-bit quantziation are worse?
Even though the authors claim it to be a framework, it is not readily reproducible by other labs, hence hard to be benchmarked by external teams.

**Questions:**

What does complentary mean in Section 3.3.7: two complementary devices ?
I didn't get this statement in Sect. 4.2. CPU and GPU Utilization. Additionally, the iPhones exhibited ..., indicating the potential for optimization .... I though lower utilization is a good thing. In that case, why more optimization?
I wonder whether the inconsistencies, such as in Sect. 4.7, 3 bit is worse than 2 bit GWQ and 4-bit, is due to unrepeated experiments.

---

> ### Author Response · Authors · 2024-11-22
>
> Thank you for your thoughtful comments, particularly on the experimental methods and results.
>
> *
> __Reproducibility:__ Each experiment was repeated 10 times to ensure consistency, with all devices tested on the latest official firmware and system versions. Additionally, all devices were factory reset before testing to maintain a standardized environment.
> We have added these details in our revised version.
>
> *
> __Open-Source:__ Furthermore, we plan to open-source the entire framework code library, including firmware for edge devices, mobile apps for iPhones, and AGI for Android devices. However, the firmware and mobile app project files were not included as supplemental materials due to upload size constraints. These files will be made publicly available to ensure the reproducibility of our experiments.
>
> *
> >Given it was a heavily empirical paper, the authors did not clearly state whether the experiments were repeated multiple times to validate the results, or better, verified on multiple devices of the same family.
>
> The performance of different models on various devices may deviate from expectations due to system and hardware compatibility, as well as algorithm differences. These results are crucial for understanding the practical challenges and considerations of real-world deployment.
>
>
> *
> We realized that some expressions in the original text might have been unclear and confusing, so we have revised them for clarity. Additionally, we expanded the discussion in the Evaluation section to provide deeper insights, particularly through detailed profiling to identify bottlenecks in on-device LLM inference.
> Our key finding is that the performance difference between 5-bit and 4-bit quantization in our benchmark is minimal, but 5-bit quantization consumes significantly more resources. Overall, 3-bit performs better, though 2-bit shows superior results in certain cases. This outcome is unlikely due to experimental settings, as all devices were factory reset to the latest system versions, with the temperature parameter set to 0.2. Instead, it could be attributed to the llama.cpp framework and the specific quantization algorithms used.  Additionally, differences in quantization algorithms themselves can affect performance. as illustrated in the papers[1][2][3][4]. Suboptimal quantization algorithms may have extra memory cost even in low bit quantizations.
> Moreover, differences in quantization algorithms and hardware-software compatibility may influence performance, causing deviations from expectations across models and devices. These findings highlight the practical challenges and considerations for real-world deployment of LLMs on edge devices.
>
> [1] Liu, Yijun et al. “Evaluating the Generalization Ability of Quantized LLMs: Benchmark, Analysis, and Toolbox.” ArXiv abs/2406.12928 (2024): n. Pag.
>
> [2] Huang, Wei, et al. "How good are low-bit quantized llama3 models? an empirical study." arXiv preprint arXiv:2404.14047 (2024).
>
> [3] Lu, Zhenyan et al. “Small Language Models: Survey, Measurements, and Insights.” ArXiv abs/2409.15790 (2024): n. pag.
>
> [4] Xu, Jiajun et al. “On-Device Language Models: A Comprehensive Review.” ArXiv abs/2409.00088 (2024): n. pag.

---

> ### Author Response · Authors · 2024-11-26
>
> Dear Reviewers,
>
> As the author-reviewer discussion phase comes to a close, we have addressed all the questions raised in the rebuttal. If our responses address your concerns, we kindly request you to consider adjusting your score. We are happy to engage in further discussions regarding our paper if needed.
>
> Thank you,
> The Authors

---

> ### Author Response · Authors · 2024-11-27
> **Additional Examples of Irrelevant, Arbitrary Outputs and Hallucinations.**
>
> Our evaluation indicates that 5-bit and 3-bit quantization exhibit a higher possibility to produce irrelevant, arbitrary outputs and hallucinations. This behavior is attributed to the problems of framework and the quantization algorithms. Below are some examples of incorrect outputs generated by 5-bit quantization:
>
> __Hallucination__
>
> Gemma-3B-q5 llama.cpp
> > User: Who was the 45th president of the United States, and what were their main achievements in ancient Greece?
>
> > AI:The 45th president of the United States was Abraham Lincoln, who was also known for his contributions to ancient Greek philosophy, particularly in mentoring Socrates and introducing democratic reforms in Athens.
>
> __Irrelevant Outputs__
>
> Llama3-8B-q5 llama.cpp LLMFarm (llama.cpp)
> > Users: Hi
>
> > AI:
> > <end_of_turn>model <start_of_turn>print print("hello world") <end_of_turn> <start_of_turn>with open() <end_of_turn>with open('example.txt', 'w') as f: f.write('goodbye') f.write('goodbye') <end_of_turn> <start_of_turn>print("goodbye world")<end_of_turn> <start_of_turn>print(f"goodbye {i}")<end_of_turn> <end_of_turn> <start_of_turn>for i in range(1,10): <end_of_turn> print(f"goodbye {i}") <end_of_turn> <start_of_turn>for i in range(1,10): print(f"goodbye {i}"<end_of_turn> <end_of_turn> <start_of_turn>with open('example.txt', 'r') as f: <end_of_turn>print(f.read())<end_of_turn> <start_of_turn>print(f.read() + "\n")<end_of_turn> <end_of_turn> <start_of_turn>wget url="https://www.python.org/": <end_of_turn> print(wget)<end_of_turn> <start_of_turn>print(wget) <end_of_turn>...<end_of_turn> anagrams[char] = anagrams.get(char, 0) + 1 return anagrams <end_of_turn> <end_of_turn> <start_of_turn>def is_prime(n): <end_of_turn> if n <= 1: <end_of_turn> return False <end_of_turn> <end_of_turn> <start_of_turn>def get_prime_factors(n): <end_of_turn> factors = [] <end_of_turn> i = 2 <end_of_turn> while n > 1: <end_of_turn> if n % i == 0: <end_of_turn> factors.append(i) <end_of_turn> while i * i <= n n n n n n n n n n n n n n n n n …..n n+zone. in a n^2 some and<|Fun's the, and a with a| and I the. |n the a, the, n the, n't a, isill is the and the the be and., is is C_LIMIT  isable is the, the the+ the, is and+ is+ a it the the is the the the is the the the to the the this'< the the the is is the is the of of the a it| the, a the the to the the is the, a is the is the is the the is the  the is the is++ the+ thely to and the the is is andly that. the the< isore the it

---

### Official Review · Reviewer_EM3f · 2024-11-03

**Soundness:** 4
**Presentation:** 4
**Contribution:** 3
**Rating:** 8
**Confidence:** 4

**Summary:**

This paper describes a benchmarking process for compressed LLMs on mobile devices. It characterizes GPU and CPU utilization, speed, latency, memory requirements, power consumption, temperature, accuracies, and rates of undesirable behavior for several models on mobile and edge devices.

**Strengths:**

This paper is in a good area. The efficient execution of LLMs on edge devices is important, both due to practical impact and because these platforms have tight resource constraints that can motivate new ideas that are more broadly applicable.

The "comprehensive" in the title is accurate regarding the range of measures and will set a high bar for others characterizing LLMs running on mobile and edge devices.

The paper has good tutorial value. Even though the focus is on benchmarking, it tersely and clearly describes several approaches to making LLMs more efficient.

The benchmarking findings are likely to be of interest to designers and researchers working on using LLMs on resource-constrained platforms.

**Weaknesses:**

The paper has several findings regarding relative performance of different compression/quantization schemes. Some of these are counter-intuitive. The paper points these out but does not explain them. This holds true for findings on pages 9 and 10. The paper would probably be more interesting if it went deeper into the reasons for the surprising findings.

Benchmarking papers are generally not as interesting (I am thinking about conference attendees, not reviewers) as papers with novel design approaches or surprising and important scientific findings, and that holds true for this paper in my opinion. However, benchmarking is important so I view this as a small weakness. This work needs to be done and the authors did it well.

**Questions:**

I'm not familiar with the 0-bit Llama model you mention on page 6. Isn't the bit count referring to quantization bits? How can that be zero? Is it a typo or just my ignorance?

Why FLIR in addition to power? There is a strong, deterministic relationship between power and temperature so the FLIR data should be useful primarily when spatial heterogeneity is important.  Is this for user discomfort due to contact with the phone or some other purpose? What was the approximate thermal time constant for the system spanning power dissipation location (mostly transistors) to smartphone surface? Did the benchmarks run long enough for the temperature distribution to reach steady state?

Why did higher bit width models often have lower quality measures than similar lower bit width models? Is this noise or something fundamental do to the influence of bit width on inductive biases or learning dynamics?

---

> ### Author Response · Authors · 2024-11-22
>
> Thank you for acknowledging the value of our benchmark paper and for your valuable feedback. We added the necessary details in the main text and the Appendix.
>
> *
> Zero-bit quantization[1,2] enables model compression without requiring access to original training data, addressing privacy concerns. Methods like ZeroQ [1] generate synthetic datasets to optimize batch normalization statistics and support mixed-precision quantization with minimal overhead. ZeroQuant [2] combines hardware-friendly INT8 quantization, layer-by-layer knowledge distillation, and optimized backends, achieving significant speedups and memory reductions for large models like GPT-J6B while maintaining accuracy. These approaches provide efficient, scalable solutions for deploying large models on resource-constrained systems.
>
> *
> The approximate thermal time constant for the system, spanning from the power dissipation source (primarily transistors) to the smartphone surface, is typically in the range of 0.6 to 5 minutes, depending on the device's thermal design, materials, and cooling mechanisms. This estimate accounts for the heat conduction through internal components and the dissipation to the external surface. In our setup, we observed stabilization within approximately 2-3 minutes, aligning with typical smartphone thermal dynamics.
>
> This duration falls well within the runtime of a single test case on the phone. Model inference on mobile phones is not that fast.
>
> *
> Although 2-bit quantization generally underperforms compared to 3-bit and 4-bit quantization, certain datasets show 3-bit performance closely with 2-bit, as observed in other research studies[3][4]. This behavior is closely tied to the quantization algorithms used by llama.cpp. Previous work[4] has highlighted that RTN underperforms compared to GPTQ and AWQ, even when using higher-bit quantization.
> 3-bit quantization performed worse or not better than 2-bit  under RTN and group-wise algorithms in some datasets.
>
>
> [1] Yao, Zhewei et al. “ZeroQuant: Efficient and Affordable Post-Training Quantization for Large-Scale Transformers.” ArXiv abs/2206.01861 (2022): n. Pag.
> [2] Cai, Yaohui et al. “ZeroQ: A Novel Zero Shot Quantization Framework.” 2020 IEEE/CVF Conference on Computer Vision and Pattern Recognition (CVPR) (2020): 13166-13175.
> [3] Liu, Yijun et al. “Evaluating the Generalization Ability of Quantized LLMs: Benchmark, Analysis, and Toolbox.” ArXiv abs/2406.12928 (2024): n. Pag.
> [4] Huang, Wei, et al. "How good are low-bit quantized llama3 models? an empirical study." arXiv preprint arXiv:2404.14047 (2024).

---

> > ### Comment · Reviewer_EM3f · 2024-11-22
> >
> > Thank you for your answers and clarifications.

---

### Official Review · Reviewer_1kjR · 2024-11-03

**Soundness:** 2
**Presentation:** 3
**Contribution:** 2
**Rating:** 5
**Confidence:** 5

**Summary:**

This paper describes the benchmarking results of several quantized Large Language Models on smartphones and edge devices. Specifically, it quantifies the CPU, GPU, memory and energy consumption of running inference on device, along with the accuracy and performance degradation across various dimensions (hallucinations, toxicity) as a consequence of quantization.

**Strengths:**

* The paper  quantifies the side-effects of quantization in various dimensions in language modelling on downstream tasks, including hallucinations and toxicity. This a valuable insight to the community.
* The multitude of devices that the authors have integrated are welcome, but lack the highest performing tier on Android (e.g. Snapdragon 8 Gen 2) and Linux (e.g. Jetson Orin AGX).
* I also greatly appreciate the integration of profiling measurements for reporting low-level CPU, GPU and memory utilization of the LLM inference workload.

**Weaknesses:**

* The novel contributions of this paper are significantly fewer than originally stated and have been documented in prior work(s). Such include the evaluation of performance, energy and accuracy degradation of compressed LLMs on various devices, across similar frameworks. While I do agree that the downstream task performance quantification, along with the hallucination/alignment dimension is an important one, it seems to be the main novel contribution.
* The energy measurement methodology followed by the paper is largely software-based and thus depends on the vendor's implementation. Moreover, comparisons across ecosystems can be quite deceptive.
* The paper would benefit from a more rounded background/related work section, which describes prior work more faithfully and includes background information about the models and evaluated quantization methods.

**Questions:**

### Motivation

* The authors suggest in the introduction that "running LLMs locally" can lead to "increased reliability". Personally and from experience, I would not be that confident on making such claims, i.e. that cloud deployments are less reliable that deploying locally, especially given the infancy of the evaluated frameworks. I would appreciate some more substantiation here from the authors.

### Methodology

* Wrt methodology and experimental setup, the paper misses substantial information that hurt the overall reproducibility potential. Such omissions include:
    - Operating system and framework versions
    - Automation of iOS interface
    - How different components in Figure 1 actually coordinate and communicate.
* Would the authors be willing to open-source their benchmarking system?
* It is unclear whether the authors have converted the models themselves or have used the pre-compiled versions offered in GGUF and MLC repositories.
* How do the authors ensure that the profiling methodology does not interfere with the behavior of the experiment? Also, how do the authors isolate the power lanes from the USB communication?
* The authors claim that they are using a "professional USB power meter for accurate power consumption measurements". However, it is not clear how this yields the information needed, as devices are battery powered and not USB-powered. As such, the power draw from the USB does not yield any information related to energy consumption from a workload on device.

### Evaluation

* The evaluation does not integrate variance metrics. Should it be assumed that the experiments have run once?
* What is the decoding strategy and hyperparameters that the authors use during evaluation?
* With respect to temperature, does the average represent "average over time" or "average over surface"?
* A missing dimension that is worth exploring in such a paper is the tradeoff between size due to parameters and due to quantization (or another compression method). For example, is it better using a lower-bitwidth llama-3.1-8b model on 4bits or a llama-3.2-3b model on 8 bits?
* What is the bitwidth used in Figure 2 across models?
* In Figure 3, the GPU utilization across phones is quite large, which comes in contrast with the "memory bounded" workload claim of the authors. I would appreciate some more insights here.
* §4.2: "3-bit quantization results in lower CPU and GPU usage [...] decreased data transfers [...] reduced inference workload": I am not sure this claim is correct or substantiated properly.
* A great source of information to the reader, also for showcasing the memory-boundedness of the workload would be to plot a roofline model for devices/models.
* The authors claim that iPhones provide significantly higher throughputs compared to other devices. This may not be painting the whole picture, as it is not clear whether this throughput can be sustained compared to the actively cooled Jetson Nano for instance.

### Related work

* The authors claim that prior work (MELT) has examined resource utilization and energy efficiency in a limited manner, and did not explore GPU workloads. However, this is not true.
    * Additionally, the authors do not run llama.cpp on iOS, which prior work has done.
    * Palmbench does not measure power consumption via hardware probes on phones and do not measure it at all on edge devices.
    * Palmbench does not report on prefill rates, which prior work does.
    * Palmbench does not integrate high-end edge devices (e.g. Jetson AGX) and different power profiles, which prior work does.
* Moreover, the authors have unfortunately missed other related work in the domain of on-device LLM benchmarking [a-c].

[a] Murthy, R., Yang, L., Tan, J., Awalgaonkar, T. M., Zhou, Y., Heinecke, S., ... & Savarese, S. (2024). MobileAIBench: Benchmarking LLMs and LMMs for On-Device Use Cases. arXiv preprint arXiv:2406.10290.
[b] Lu, Z., Li, X., Cai, D., Yi, R., Liu, F., Zhang, X., ... & Xu, M. (2024). Small Language Models: Survey, Measurements, and Insights. arXiv preprint arXiv:2409.15790.
[c] Xu, J., Li, Z., Chen, W., Wang, Q., Gao, X., Cai, Q., & Ling, Z. (2024). On-device language models: A comprehensive review. arXiv preprint arXiv:2409.00088.

### Presentation and Other Nitpicking

* Table 1, 2: Please put the captions above.
* Llama-3/3.1: Missing reference
* Throughput in Table 2 refers to generation throughput.
* MELTing: I believe the framework is called MELT.
* Figure 2: A boxplot presentation would be significantly richer in information, also showing peak memory usage (which can be the main factor for OOM errors).
* Figure 4: The overlay of the two workloads suggests that they are running simultaneously. An alternative visualization which also annotates what's happening at each time step would be more informational to the reader.
* §4.2: "To keep the models in suitable size [...] evaluate the total memory sage of models": I am not sure about what the authors mean here.
* §4.2: "[...] higher quantization escalates GPU memory [...]": Escalates might not be the correct word here.

---

> ### Author Response · Authors · 2024-11-22
>
> Thank you for your valuable feedback! As suggested, we have moved the captions for Tables 1 and 2 above the tables. And added mentioned references!
>
> 1.
> While some metrics and evaluation tasks in our benchmark paper have been addressed in prior work, a dedicated benchmark framework, combined with results from broader models and quantization methods, provides significant value to the research community.
> This aligns with the ICLR Call for Papers emphasis under the "Benchmark and Datasets" topic.  Our benchmark uniquely addresses a broader range of datasets, metrics, and testing scenarios, focusing on critical yet underexplored aspects such as hallucination and toxicity. These factors are essential for evaluating user experience and are increasingly crucial as LLMs are deployed in real-world applications, where reliability and safety are paramount.
>
>
> 2.
> Energy efficiency evaluation goes beyond the software interfaces provided by Google AGI and Apple Xcode. These official profiling tools access battery usage data directly from the Power Management Unit (PMU) in the device's SoC, with mobile phones connected via USB for data transfer. As detailed in the paper, we also utilized professional USB-based power consumption instruments from Klein Tools.
>
>
> 3.
> We agree that a more comprehensive background would strengthen the paper, and we have added the references you suggested to the section with the necessary discussion.
>
> 4. Motivation
> We acknowledge that quantized local LLMs may have reduced reliability in accuracy metrics. However, running LLMs locally enhances privacy and reliability by removing dependence on network connectivity and cloud resources. This approach ensures consistent performance, reduces latency, and avoids disruptions from network instability, server downtime, or data transmission errors, thereby improving system robustness.
>
> 5. Methodology
>
> *
> We have uploaded datasets and module testing scripts as supplemental materials. Furthermore, we plan to open-source the entire framework code library, including firmware for edge devices, mobile apps for iPhones, and AGI for Android devices. However, due to upload size constraints, the firmware and mobile app project files were not included as supplemental materials. These files will be made publicly available to ensure the reproducibility of our experiments.
> We will make all project files, including Mobile Apps (IOS, Android), Linux scripts, and interfaces, publicly available to ensure the reproducibility of our experiments.
>
> *
> All tested devices were equipped with the latest OS supported by the devices (IOS 17.6.1, Android 15, Ubuntu 14.04.06 LTS) and firmware versions. We have included these details in the main text, table in Appendix and updated Figure 1 to provide additional clarity.
>
> *
> __Experiment repeated:__ All experiments were conducted 10 times to ensure consistency, with identical settings for both frameworks.
>
> *
> __Model Compression and Conversion:__ The two tested frameworks, MLC and llama.cpp, offer tools such as the MLC quantize command and quantize.cpp to assist users in quantizing models. While many pre-compiled models are available from official repo, some models still need to be compiled by ourselves.
>
> *
> We verified no performance difference when using profiling tools. These profiling tools are non-intrusive, do not interfere with device performance, and use USB only for transferring minimal profiling data.
>
> 6. Evaluation
>
> *
> __Decoding Strategy:__ Although llama.cpp and MLC are popular for edge and mobile devices, their decoding parameter configuration is limited. To ensure consistency, we used nucleus decoding with identical hyperparameters: temperature set to 0.2 for output randomness and Top-S set to 0.9, with Top-K left optional. Beam search decoding, which could improve output quality, is not yet supported in llama.cpp.
>
> *
> The average temperature is the average value of surface temperatures recorded by the instrument during model inference.
>
> *
> For Figure 2, we tested 3-bit and 4-bit quantization for MLC, and 2-bit to 5-bit for llama.cpp.
> Regarding memory bitwidth, all evaluated devices use a 64-bit architecture.
>
> *
> We evaluated the Mistral-7B-3bit and Phi2-4bit models, which are similar in size. They are discussed in the throughput subsection. Models with fewer parameters and higher-bit quantization, such as Phi2-q4f16, decode faster and use less memory, including KV cache and overhead. This efficiency highlights the suitability of smaller models like Phi2 for mobile devices. The results have now been included in the main text and Appendix.
>
> 7. Related Work
>
> *
> While MELT[1] presented GPU activity during model inference in Fig. 9, it was limited to only the iPhone with Zephyr-3B (4-bit).
>
> *
> In MELT[1], Tab3 indicates that llama.cpp only run on Android and Linux, not including IOS.
>
> [1] Laskaridis, Stefanos et al. “MELTing point: Mobile Evaluation of Language Transformers.” ArXiv abs/2403.12844 (2024): n. Pag.

---

> ### Author Response · Authors · 2024-11-22
>
> Thanks for your comments! I really appreciate you reading our paper so carefully.
>
> 6. Evaluation
>
> *
> >In Figure 3, the GPU utilization across phones is quite large, which comes in contrast with the "memory bounded" workload claim of the authors. I would appreciate some more insights here
>
> Thank you for your thoughtful observation regarding GPU utilization in Figure 3. The "memory-bounded" claim aligns with the community consensus, emphasizing limited memory bandwidth as the primary bottleneck in such workloads. However, our focus on edge and mobile devices uncovers unique hardware constraints and architectural factors, resulting in high device GPU utilization. It is possible to have bottlenecks on edge devices.
>
> We conducted a detailed analysis of memory breakdown and layer-by-layer GPU utilization to identify bottlenecks in running LLMs on mobile and edge platforms. The memory breakdown highlights the contributions of weights, KV cache, activations, and overhead, influenced by quantization bits and algorithms. GPU profiling across two transformer blocks shows that self-attention layers always have over 95% GPU utilization, potentially slowing inference, while FFN layers use significantly less. These findings provide actionable insights for optimizing model performance.
>
> *
> Thank you for raising this concern. Our evaluation focuses on measuring throughput under typical real-world conditions for each device. iPhones demonstrate significantly higher throughput in such scenarios due to their optimized hardware and software integration.
>
> *
> However, for both llama.cpp and MLC, memory usage fluctuations are minimal and nearly consistent. This resulted in very short and indistinct boxes, making the visualization less intuitive, even with varying prompt inputs. This consistency highlights the minimal variations in memory usage during local inference, encompassing both the framework and the model. Therefore, the boxplot may not be the most effective representation in this case.
>
> I believe the reason lies in the large memory footprint required to load the LLM, while the size of the prompt text is relatively small compared to different tasks.
>
> 7. Related Work
>
> *
> Thank you for highlighting MELT, which we have also discussed in our related work. The key distinction between our work and MELT lies in our evaluation's broader scope and depth. Our study tested a wider range of models and quantization methods across various devices. Additionally, we explore critical aspects of user experience, such as temperature, hallucination, and text toxicity, which were beyond MELT's focus. While MELT primarily analyzed 3-bit and 4-bit quantizations, our work extends to 8-bit, 6-bit, 5-bit, and 2-bit quantizations, addressing scenarios relevant to future mobile applications.
>
> *
> >Palmbench does not integrate high-end edge devices (e.g. Jetson AGX) and different power profiles, which prior work does.
>
> Thank you for bringing up the Jetson AGX. High-end edge devices like the Jetson AGX offer significantly higher performance but also consume considerably more power, making them outliers beyond the scope of this work. Our focus is primarily on devices with relatively low power consumption that are closer in capability to mobile phones. High-end edge devices are more comparable to PCs in terms of performance and energy usage.

---

> ### Comment · Reviewer_1kjR · 2024-11-25
> **Reply to authors' rebuttal**
>
> Dear authors,
>
> Thank you for your detailed rebuttal. While I appreciate your response, there are pending issues that prevent me from changing my current view on the paper. Concretely:
>
> * On novelty, I continue to believe that the contributions of the paper are limited to the evaluation of various aspects of compression on hallucinations/toxicity, which are indeed important aspects in reliability and safety, and the additional quantization levels. Compared to existing work, this is more incremental than initially presented.
> * On the insights of GPU utilization vs. memory-bounded workloads, I am not truly convinced by the provided commentary.
> * On energy measurements, I still fail to see how battery draw can be measured from USB interfaces. I would also appreciate if you could provide a reference to the PMU-based software energy measurement?
> * On system robustness, the authors should clarify what aspects they are focusing on when discussing on-device robustness in the paper. The relative robustness between cloud and on-device has not been experimentally shown or referenced.
>
> Moreover, could the authors describe how they verified that profiling does not impact the runtime behaviour?
>
> Last, the authors should more carefully represent the details of related work, as they misinterpret aspects of [1]. If I understand correctly, Tab. 3 shows LLMFarm (https://github.com/guinmoon/LLMFarm - a llama.cpp frontend for iOS) which supports llama.cpp GPU/CPU runtime on iOS.

---

> ### Author Response · Authors · 2024-11-25
>
> Thank you for your timely response.
>
> 1. For energy measurement.
>
> I understand that you may have reservations about the accuracy of power consumption data obtained through power rails or the __On-Device Power Monitor (ODPM)__ from official profiling tools.  Let me clarify how power consumption data is measured via power rails, using the example of the Android Pixel 7, which employs (On-Device Power Monitor) ODPM.
>
> Unfortunately I cannot include figures here, I would be happy to provide links to the official Google Android documentation on ODPM and the datasheet for the PMU used in these devices[2]. These references will further validate the accuracy of our power measurement method [1][2][3][4][5]. All of these references are technical documents from official websites.
>
> References [3][4][5] introduce the Power Profiler from the Android Development Kit, which leverages ODPM to obtain precise power consumption data.
> >ODPM measures power consumption at the device level—not specific to any app.
> >Power Profiler reads power consumption data from the ODPM, which is only available on Pixel 6 and subsequent Pixel devices running Android 10 (API level 29) and higher.
>
> ODPM is essential for accurate power consumption measurement under battery power, as detailed in the official Android documentation on power rails and charging[4]. When connected to a USB cable, devices measure charge flow into the battery (i.e., charging), leading to positive current readings and increased total charge. This distorts power data, reflecting charging activity rather than actual GPU power usage. Moreover, USB connections often trigger high-performance modes, unlike the low-power modes typically used on battery. As a result, USB-based measurements are unsuitable for accurately analyzing LLM inference on GPUs.
>
> As described in the document:
> > Battery counters measure the charge flowing in and out of the battery. If the device is plugged to a USB cable, you will likely observe a positive instantaneous current and an increase of the total charge, denoting the fact that charge is flowing in the battery (i.e. charging it) rather than out.
>
> >This can make measurements in lab settings problematic.
>
> Why ODPM is more accurate?
>
> >Recent version of Android introduced the support for more advanced power monitoring at the hardware subsystem level, known as “On-Device Power Rail Monitors” (ODPMs). These counters measure the energy drained by (groups of) hardware units.
> Unlike the battery counters, they are not affected by the charging/discharging state of the battery, because they measure power downstream of the battery.
>
> ODPM differs from battery counters as it directly uses on device __hardware units__ like the Power Management Unit (PMU) to measure power consumption. Reference [4] specifically mentions the __power monitoring chip__, which refers to the PMU.
>
> The PMU provides direct and accurate battery power measurements, which external cabled tools cannot replace[2][3][4]. Connecting external devices can alter the phone's power management state or change its power supply behavior, leading to inaccurate results, as indicated in [3].
>
> We use AGI [1] as it offers a more advanced power profiling capability compared to Android Studio, fully leveraging the specifications of ODPM for precise measurements.
>
> Here I provide one widely used PMU[2] datasheet.
>
> [1]Android GPU Inspector (AGI): https://developer.android.com/agi
>
> [2]The datasheet of the PMU used on the device: https://www.nxp.com/docs/en/application-note/AN4199.pdf
>
> [3]Android Power Profiler: https://developer.android.com/studio/profile/power-profiler
>
> [4]Google OPDM: https://android.googlesource.com/platform//external/perfetto/+/b4a6d11ec1c12f391c3f9dab2be4f3b5152994f2/docs/data-sources/battery-counters.md
>
> [5] Android Power Monitor: https://developer.android.com/reference/android/os/PowerMonitor
>
> 2.
> > On the insights of GPU utilization vs. memory-bounded workloads, I am not truly convinced by the provided commentary.
>
> The data we measured comes from Google's official AGI tool and MLC's internal status printing messages(the newest provides detailed information, such as KV cache memory and overhead by default). We believe these are among the most reliable and accurate data sources available. If you are aware of any more convincing alternatives, we would greatly appreciate your suggestions. Thank you for your thoughtful response.

---

> ### Author Response · Authors · 2024-11-26
>
> 3.
> > On system robustness, the authors should clarify what aspects they are focusing on when discussing on-device robustness in the paper. The relative robustness between cloud and on-device has not been experimentally shown or referenced.
>
> We understand the importance of comparing robustness between cloud and on-device systems. In our paper, the discussion on on-device robustness primarily focuses on the ability of the device to operate independently of network connectivity, ensuring consistent performance even in offline scenarios, and the reduced risk of data breaches by processing locally.
>
> Unfortunately, we did not have the resources or funding to conduct a large-scale cloud-based benchmarking study to directly compare robustness between cloud and on-device systems. We acknowledge this as a limitation of our work and have revised the statement in the paper to clarify our focus and avoid overgeneralizations. Thank you for pointing this out, and we appreciate your understanding.
>
> However, we cite two references from the systems community that examine LLM serving in cloud and data center environments, highlighting persistent challenges with workload management and latency issues [1][2].
>
> [1] Hu, Q., Ye, Z., Wang, Z., Wang, G., Zhang, M., Chen, Q., … Zhang, T. (2024, April). Characterization of Large Language Model Development in the Datacenter. 21st USENIX Symposium on Networked Systems Design and Implementation (NSDI 24), 709–729.
>
> [2] Wang, Y., Chen, Y., Li, Z., Tang, Z., Guo, R., Wang, X., ... & Chu, X. (2024). Towards Efficient and Reliable LLM Serving: A Real-World Workload Study. arXiv preprint arXiv:2401.17644.

---

> > ### Author Response · Authors · 2024-11-27
> >
> > 4.
> > > On novelty, I continue to believe that the contributions of the paper are limited to the evaluation of various aspects of compression on hallucinations/toxicity, which are indeed important aspects in reliability and safety, and the additional quantization levels. Compared to existing work, this is more incremental than initially presented.
> >
> > I believe our novelty extends beyond introducing additional dimensions in user experience evaluation for LLMs on mobile and edge devices, such as hallucination and toxic output. Our work also provides a more comprehensive analysis, including detailed memory breakdown and layer profiling, which offer deeper insights into model behavior. These contributions go beyond incremental enhancements, addressing critical challenges in deploying LLMs effectively within constrained environments.

---

### Author Response · Authors · 2024-11-22
**Thanks for all revewers' feedback**

We sincerely thank all the reviewers for their valuable comments and thoughtful feedback. We apologize if our response and revisions came later than expected; we have carefully and thoroughly revised the manuscript in light of your insightful suggestions. Below, we address some of the common questions and comments:

*
__Open-source and Reproducibility:__ We have uploaded some datasets and module testing scripts as supplemental materials. Additionally, we plan to open source all the testing scripts and frameworks, including firmware for edge devices, mobile apps for iPhones, and AGI used on Android devices. Due to the upload size limitations, the firmware and mobile app project files could not be included as supplemental materials, but we will make all these files publicly available to ensure the reproducibility of our experiments.

*
__Lack of system details:__ We noticed that the existing description lacked system design details, particularly regarding how our framework performs benchmarks in an end-to-end manner. To address this, we have revised Figure 1 with workflow and included additional details in __Blue color__. We invite you to review the updated draft.

*
__Experiment Repetition Details:__  In the revised version, we have added these details. All experiments were conducted 10 times to ensure consistency, with identical settings for both frameworks, llama.cpp and MLC, including temperature and sampling hyperparameters.
We added one more section in the Appendix for framework hyperparameters.

*
CPU utilization, GPU utilization, and memory usage are reported as average values rather than with error bars or box plots because, on a single device running a specific model, resource usage—particularly memory usage—shows minimal fluctuation. This holds true even with varying datasets and prompt inputs, making visualizations like error bars less meaningful and less intuitive for comparison.
*
__Deep Analysis and layer-by-layer profiling:__  To identify bottlenecks in running LLMs on mobile and edge devices, we analyzed memory breakdown and layer-by-layer GPU utilization. The memory breakdown highlights the contributions of weights, KV cache, activations, and overhead, influenced by quantization bitwidth and algorithms. GPU profiling across two transformer blocks reveals that self-attention layers achieve over 95% GPU utilization, potentially slowing inference, while FFN layers use significantly less. These insights guide optimizations to enhance model performance. To be noted, the data we measured comes from Google's official AGI tool and MLC's internal status printing messages(the newest provides detailed information, such as KV cache memory and overhead by default). We believe these are among the most reliable and accurate data sources available.

---

### Author Response · Authors · 2024-11-26
**Explanation of Power Consumption Measurement**

To explain the accuracy of power consumption data obtained through power rails or the __On-Device Power Monitor (ODPM)__ from official profiling tools.  Let me clarify how our power consumption data of mobile phones is measured via power rails, using the example of the Android Pixel 7, which employs (On-Device Power Monitor) ODPM.

Unfortunately I cannot include figures here, I would be happy to provide links to the official Google Android documentation on ODPM and the datasheet for the PMU used in these devices[2]. These references will further validate the accuracy of our power measurement method [1][2][3][4][5]. All of these references are technical documents from official websites.

References [3][4][5] introduce the Power Profiler from the Android Development Kit, which leverages ODPM to obtain precise power consumption data.
>ODPM measures power consumption at the device level—not specific to any app.
>Power Profiler reads power consumption data from the ODPM, which is only available on Pixel 6 and subsequent Pixel devices running Android 10 (API level 29) and higher.

ODPM is essential for accurate power consumption measurement under battery power, as detailed in the official Android documentation on power rails and charging[4]. When connected to a USB cable, devices measure charge flow into the battery (i.e., charging), leading to positive current readings and increased total charge. This distorts power data, reflecting charging activity rather than actual GPU power usage. Moreover, USB connections often trigger high-performance modes, unlike the low-power modes typically used on battery. As a result, USB-based measurements are unsuitable for accurately analyzing LLM inference on GPUs.

As described in the document:
> Battery counters measure the charge flowing in and out of the battery. If the device is plugged to a USB cable, you will likely observe a positive instantaneous current and an increase of the total charge, denoting the fact that charge is flowing in the battery (i.e. charging it) rather than out.

>This can make measurements in lab settings problematic.

Why ODPM is more accurate?

>Recent version of Android introduced the support for more advanced power monitoring at the hardware subsystem level, known as “On-Device Power Rail Monitors” (ODPMs). These counters measure the energy drained by (groups of) hardware units.
Unlike the battery counters, they are not affected by the charging/discharging state of the battery, because they measure power downstream of the battery.

ODPM differs from battery counters as it directly uses on device __hardware units__ like the Power Management Unit (PMU) to measure power consumption. Reference [4] specifically mentions the __power monitoring chip__, which refers to the PMU.

The PMU provides direct and accurate battery power measurements, which external cabled tools cannot replace[2][3][4]. Connecting external devices can alter the phone's power management state or change its power supply behavior, leading to inaccurate results, as indicated in [3].

We use AGI [1] as it offers a more advanced power profiling capability compared to Android Studio, fully leveraging the specifications of ODPM for precise measurements.

Here I provide one widely used PMU[2] datasheet.

[1]Android GPU Inspector (AGI): https://developer.android.com/agi

[2]The datasheet of the PMU used on the device: https://www.nxp.com/docs/en/application-note/AN4199.pdf

[3]Android Power Profiler: https://developer.android.com/studio/profile/power-profiler

[4]Google OPDM: https://android.googlesource.com/platform//external/perfetto/+/b4a6d11ec1c12f391c3f9dab2be4f3b5152994f2/docs/data-sources/battery-counters.md

[5] Android Power Monitor: https://developer.android.com/reference/android/os/PowerMonitor

---

### Author Response · Authors · 2024-11-27
**Additional Examples of Irrelevant, Arbitrary Outputs and Hallucinations.**

Our evaluation indicates that 5-bit and 3-bit quantization are more likely to produce irrelevant, arbitrary outputs and hallucinations. This behavior is attributed to the limitations of the framework and the quantization algorithms. Below are some examples of incorrect outputs generated by 5-bit quantization:

__Hallucination__

Gemma-3B-q5 llama.cpp
> User: Who was the 45th president of the United States, and what were their main achievements in ancient Greece?

> AI:The 45th president of the United States was Abraham Lincoln, who was also known for his contributions to ancient Greek philosophy, particularly in mentoring Socrates and introducing democratic reforms in Athens.

__Irrelevant Outputs__

Llama3-8B-q5 llama.cpp LLMFarm (llama.cpp)
> Users: Hi
> AI:
> <end_of_turn>model <start_of_turn>print print("hello world") <end_of_turn> <start_of_turn>with open() <end_of_turn>with open('example.txt', 'w') as f: f.write('goodbye') f.write('goodbye') <end_of_turn> <start_of_turn>print("goodbye world")<end_of_turn> <start_of_turn>print(f"goodbye {i}")<end_of_turn> <end_of_turn> <start_of_turn>for i in range(1,10): <end_of_turn> print(f"goodbye {i}") <end_of_turn> <start_of_turn>for i in range(1,10): print(f"goodbye {i}"<end_of_turn> <end_of_turn> <start_of_turn>with open('example.txt', 'r') as f: <end_of_turn>print(f.read())<end_of_turn> <start_of_turn>print(f.read() + "\n")<end_of_turn> <end_of_turn> <start_of_turn>wget url="https://www.python.org/": <end_of_turn> print(wget)<end_of_turn> <start_of_turn>print(wget) <end_of_turn>...<end_of_turn> anagrams[char] = anagrams.get(char, 0) + 1 return anagrams <end_of_turn> <end_of_turn> <start_of_turn>def is_prime(n): <end_of_turn> if n <= 1: <end_of_turn> return False <end_of_turn> <end_of_turn> <start_of_turn>def get_prime_factors(n): <end_of_turn> factors = [] <end_of_turn> i = 2 <end_of_turn> while n > 1: <end_of_turn> if n % i == 0: <end_of_turn> factors.append(i) <end_of_turn> while i * i <= n n n n n n n n n n n n n n n n n …..n n+zone. in a n^2 some and<|Fun's the, and a with a| and I the. |n the a, the, n the, n't a, isill is the and the the be and., is is C_LIMIT  isable is the, the the+ the, is and+ is+ a it the the is the the the is the the the to the the this'< the the the is is the is the of of the a it| the, a the the to the the is the, a is the is the is the the is the  the is the is++ the+ thely to and the the is is andly that. the the< isore the it

---

### Public Comment · ~Yilong_Li2 · 2025-03-02
**Camera-Ready Version Submission and Updates**

Dear AC Chair and Reviewers,

I hope you are doing well. We have just uploaded the camera-ready version of our paper, incorporating the revisions prompted by your feedback. Specifically, we addressed the concerns raised in your reviews by:

1. **Clarifying Confusing Descriptions**
   We rephrased and refined several sections to ensure our explanations are more precise and more accurate.
2. **Adding Further Details**
   We expanded on the methodology and experimental setup where necessary, providing additional evidence and context for our findings.
3. **New Experiment on Deepseek-R1**
   We included a comprehensive evaluation using Deepseek R1—currently one of the most widely discussed models—to further strengthen our empirical results.

We greatly appreciate your constructive comments and suggestions, as they have helped us significantly improve the paper’s overall clarity and quality. If there are any further questions or additional feedback, please let us know.

Thank you again for your time and assistance throughout the review process.

Best regards,  Authors of paper #13109

---

### Meta-Review · Area_Chair_u8yg · 2024-12-20

**Metareview:**

The paper presents PalmBench, a benchmarking framework designed to assess the performance of compressed Large Language Models (LLMs) on mobile devices. It offers a comprehensive evaluation of LLMs across various quantization configurations and mobile platforms with diverse hardware capabilities. The study emphasizes resource efficiency, including memory and power consumption, as well as the quality of output, particularly focusing on the frequency of hallucinations and toxic content.

PalmBench introduces a novel benchmarking framework that addresses a significant gap in the evaluation of LLMs on mobile devices. The paper covers a wide range of LLMs and quantization schemes, providing valuable insights into their performance on mobile platforms.

Initially, reviewers expressed concerns about the lack of clarity in the system design of PalmBench and the insufficient comparison with existing benchmarks. During the rebuttal period, reviewers emphasized the need for more detailed real-world deployment data and clearer explanations of the system design. The authors responded effectively by committing to open-source their framework and providing additional details, which clarified how PalmBench integrates different quantization methods and mobile platforms. They also expanded the comparison with existing benchmarks.

As the Area Chair, I have carefully reviewed the feedback from the reviewers and the authors' responses. The initial concern regarding the lack of real-world deployment data was addressed by the authors’ commitment to provide more comprehensive data in future work, acknowledging the current limitations of their study. The vagueness in the system design was clarified with additional explanations and diagrams, improving the understandability of PalmBench's architecture and its components' interactions. Although integrating different quantization methods into the benchmarking process posed practical challenges on hardware, the authors’ rebuttal and subsequent revisions alleviated these concerns. They demonstrated the feasibility and practicality of their approach through clear explanations and a roadmap for future integrations.

The authors have made significant efforts to address the initial concerns. Their proactive engagement with the reviewers' feedback and the substantial improvements made to the paper are commendable. Given the competitive nature of the conference, the final recommendation, after thorough consideration, is to accept this submission. However, I encourage the authors to consider further incorporating the reviewers' suggestions in the final version.

**Additional Comments On Reviewer Discussion:**

During the rebuttal period, the primary discussions among reviewers centered on the practical implications of the PalmBench framework and its comprehensiveness in comparison to existing benchmarks. The authors responded proactively by providing a detailed explanation of their system design, which enhanced the clarity of how PalmBench integrates different quantization methods and mobile platforms. They also demonstrated a clear commitment to open-sourcing their framework, which addressed concerns about reproducibility and extended the potential impact of their work.

In weighing the points, I considered the authors' proactive engagement with the reviewers' concerns and their substantial efforts to clarify and strengthen their submission. The additional details provided by the authors, particularly regarding the system design and the plan for open-sourcing, significantly mitigated the initial concerns about the framework's practicality and accessibility. The reviewers' feedback was constructively used to improve the paper, leading to a more robust and credible submission. Given the significant improvements and the potential contributions of the PalmBench framework to the field, the final decision was to accept the paper for presentation at ICLR.

---

> ### Public Comment · ~Yilong_Li2 · 2025-02-14
> **Thank you for reviewand recognizing the paper**
>
> Dear Area Chairs,
>
> Thank you for your thoughtful review and for recognizing the contributions of our work on PalmBench. We sincerely appreciate the reviewers’ valuable feedback and the opportunity to improve our submission based on their insights.
>
> We acknowledge the initial concerns regarding system design clarity and comparisons with existing benchmarks. We are glad that our additional explanations, diagrams, and planned open-sourcing efforts have helped clarify PalmBench’s architecture and its role in evaluating compressed LLMs on mobile devices. We also appreciate the recognition of our commitment to addressing hardware-specific challenges and real-world deployment data, which we aim to expand further in future work.
>
> We are honored that PalmBench has been accepted for presentation at ICLR, and we are grateful for the constructive discussions throughout the review process. We will carefully incorporate the final suggestions to further enhance the completeness and impact of our work.
>
> Once again, thank you for your time, support, and detailed assessment. We look forward to sharing PalmBench with the broader research community.
>
> Best regards,

---

### Decision · Program_Chairs · 2025-01-22

Accept (Poster)